# QUEST: A ROBUST ATTENTION FORMULATION USING QUERY-MODULATED SPHERICAL ATTENTION

**Hariprasath Govindarajan**[1,2]   **Per Sidén**[2]   **Jacob Roll**[2]   **Fredrik Lindsten**[1]
[1]Linköping University, Sweden   [2] Qualcomm Auto Ltd Sweden Filial
{hargov,psiden,jroll}@qti.qualcomm.com,
fredrik.lindsten@liu.se

## ABSTRACT

The Transformer model architecture has become one of the most widely used in deep learning and the attention mechanism is at its core. The standard attention formulation uses a softmax operation applied to a scaled dot product between query and key vectors. We explore the role played by norms of the queries and keys, which can cause training instabilities when they arbitrarily increase. We demonstrate how this can happen even in simple Transformer models, in the presence of easy-to-learn spurious patterns in the data. We propose a new attention formulation, QUEry-modulated Spherical aTtention (QUEST), that constrains the keys to a hyperspherical latent space, while still allowing individual tokens to flexibly control the sharpness of the attention distribution. QUEST can be easily used as a drop-in replacement for standard attention. We focus on vision applications while also exploring other domains to highlight the method's generality. We show that (1) QUEST trains without instabilities and (2) produces models with improved performance (3) that are robust to data corruptions and adversarial attacks.

## 1 INTRODUCTION

Transformers (Vaswani et al., 2017) are one of the most widely used model architectures across many domains in recent times. Each domain has adapted Transformers to build domain-specific variants such as Vision Transformers (ViT) (Dosovitskiy et al., 2020) in computer vision, GPT (Radford et al., 2018) in natural language processing, PointTransformer (Zhao et al., 2021) for 3D pointcloud data and Conformer (Gulati et al., 2020) in speech recognition, to name a few. A core building block of the Transformer that is common across such variants is the attention mechanism (Bahdanau et al., 2015; Britz et al., 2017; Luong et al., 2015). Although several different variants of the Transformer have been developed, they commonly use the vanilla attention mechanism consisting of a scaled dot-product followed by a softmax operation. Despite their success, training Transformer models can be challenging due to training instabilities (Chowdhery et al., 2023; Dehghani et al., 2023; Li et al., 2022a; Wortsman et al., 2024; Davis et al., 2021; Liu et al., 2020b; Zhai et al., 2023). Many training techniques consisting of initialization methods (Kedia et al., 2024; Huang et al., 2020), specific hyperparameter schedules (Liang et al., 2022; Kobayashi et al., 2024), normalizations (Dehghani et al., 2023; Wang et al., 2019; Xiong et al., 2020) and optimization strategies (Qi et al., 2025) have been developed to mitigate these issues. While this has limited such training instabilities to a great extent, why this issue occurs is still not fully understood (Hajra, 2025).

We study the scaled dot-product attention formulation and identify the roles of different components of this attention. We find that the arbitrary vector norms of the queries and keys may potentially cause the exploding attention logits, that is known to cause training instabilities (Zhai et al., 2023; Dehghani et al., 2023). Through a toy example, we demonstrate a scenario where the norms of these vectors increase and can lead to the model being stuck at a suboptimal solution. Even in stably trained models, we show that the model can concentrate attention on a few tokens instead of relying on all relevant tokens (see Figure 1 for an example using a ViT). We propose a new formulation, called Query-modulated Spherical Attention (QUEST), that displays improved training robustness across different hyperparameters. QUEST is a very simple modification of the standard attention and can easily be used as a drop-in replacement in any Transformer. Nevertheless, through extensive experiments

on Transformers used in different domains, we show that our proposed attention formulation can bring consistent performance gains by learning robust patterns in the data. This is further reflected in improved robustness to adversarial attacks and data corruptions.

## 2 ATTENTION, PLEASE!

The attention mechanism is the core component of Transformers and operates on set-like or sequential data. In this section, we first present a background on scaled dot product attention (SDPA), which is the most widely used formulation of attention. Then, we provide an interpretation of the different components of this form of attention. Using that motivation, we propose a new attention formulation and finally, we use a toy example to demonstrate how it improves over SDPA and other related attention variants.

**Scaled dot product attention:** In the self-attention paradigm, the attention mechanism[1] transforms an input sequence (of length $N$) $\boldsymbol{X} \in \mathbb{R}^{N \times D}$ to an output sequence $\boldsymbol{Z} \in \mathbb{R}^{N \times D}$. Each item in this sequence is referred to as a token. Intuitively, this can be viewed as a process of relating each input token to the other tokens and aggregating some relevant information from these related tokens. In multi-head self-attention, this is repeated over several heads, to obtain $H$ different outputs $\boldsymbol{Z}_h \in \mathbb{R}^{N \times D_H}$, where $D_H = D/H$. The output $\boldsymbol{Z}_h$ is obtained as:

$$\boldsymbol{Z}_h = \boldsymbol{A}_h \boldsymbol{V}_h = \text{softmax}\left(C\boldsymbol{Q}_h \boldsymbol{K}_h^T\right) \boldsymbol{V}_h$$

where the queries $\boldsymbol{Q}_h$, keys $\boldsymbol{K}_h$ and values $\boldsymbol{V}_h$ for head $h$ are obtained as $\boldsymbol{Q}_h = \boldsymbol{X}\boldsymbol{W}_{Q,h}^T$, $\boldsymbol{K}_h = \boldsymbol{X}\boldsymbol{W}_{K,h}^T$ and $\boldsymbol{V}_h = \boldsymbol{X}\boldsymbol{W}_{V,h}^T$ respectively. The softmax is applied row-wise to the matrix $C\boldsymbol{Q}_h \boldsymbol{K}_h^T$. Typically, the scaling factor is a constant, $C = 1/\sqrt{D_H}$. For the sake of brevity, we will ignore the head index $h$ from the notation henceforth and denote vanilla attention as: $\boldsymbol{A} = \text{softmax}\left(C\boldsymbol{Q}\boldsymbol{K}^T\right)$.

Figure 1: Class-activation maps for an image from the Macaw class in ImageNet, generated using AG-CAM (Leem & Seo, 2024). Standard attention concentrates on few bird instances (see first row) and mis-classifies the image if the region containing those instances is noised (see third row). This indicates that the birds in the bottom half of the image do not contribute to the correct prediction in standard attention. Hence, when the top part of the image is noised, the model focuses on the birds in the bottom part of the image since they are the most salient object in the image then, but results in a misclassification. QUEST attention attends evenly to different bird instances and classifies the image correctly even if some of the bird instances are noised. A more diverse attention can make the models more robust to input data variations, which can be observed in the improved model robustness in 4.1.2.

### 2.1 AN ALTERNATIVE VIEW ON ATTENTION

Let $\boldsymbol{v} = \|\boldsymbol{v}\|\bar{\boldsymbol{v}}$ where $\|\boldsymbol{v}\|$ is the norm of the vector and $\bar{\boldsymbol{v}}$ is a unit vector. The attention corresponding to token $i$ can be written as:

$$\boldsymbol{A}_i = \text{softmax}\left(C\boldsymbol{q}_i \boldsymbol{K}^T\right) = \text{softmax}\left(C\|\boldsymbol{q}_i\|\bar{\boldsymbol{q}}_i \boldsymbol{K}^T\right)$$

$$= \left\{ \frac{\exp[C\|\boldsymbol{q}_i\|\|\boldsymbol{k}_1\|(\bar{\boldsymbol{q}}_i \cdot \bar{\boldsymbol{k}}_1)]}{\sum_{j'=1}^N \exp[C\|\boldsymbol{q}_i\|\|\boldsymbol{k}_{j'}\|(\bar{\boldsymbol{q}}_i \cdot \bar{\boldsymbol{k}}_{j'})]}, ..., \frac{\exp[C\|\boldsymbol{q}_i\|\|\boldsymbol{k}_N\|(\bar{\boldsymbol{q}}_i \cdot \bar{\boldsymbol{k}}_N)]}{\sum_{j'=1}^N \exp[C\|\boldsymbol{q}_i\|\|\boldsymbol{k}_{j'}\|(\bar{\boldsymbol{q}}_i \cdot \bar{\boldsymbol{k}}_{j'})]} \right\} \quad (1)$$

---

[1]Although we focused on self-attention, the interpretation in this section and the proposed QUEST attention are also applicable to other attention paradigms like cross-attention. We explore this in A.5.1.

It can be noted that the norms of the queries and keys perform distinct roles in attention. The query norm $\|\boldsymbol{q}_i\|$ scales all the attention logits and thus, controls the sharpness of the attention distribution for that token. A higher query norm results in a sharper attention and focuses on fewer tokens whereas a smaller query norm results in a softer distribution, aggregating information from a larger number of tokens. The dot product $(\bar{\boldsymbol{q}}_i \cdot \bar{\boldsymbol{k}}_j)$ denotes the similarity or vector alignment between each pair of query and key tokens. The key norm $\|\boldsymbol{k}_j\|$ controls the contribution to "general" attention from the key $j$. For a query token that is uniformly distributed in the query-key latent space, the queries are more likely to attend to key tokens that have higher norms. The token that gets the most attention from a query token depends on a combination of its vector alignment and its key norm, $\|\boldsymbol{k}_j\|(\bar{\boldsymbol{q}}_i \cdot \bar{\boldsymbol{k}}_j)$.

During training, if the information from a token helps reduce the loss objective, its key norm and the query norm of the token (e.g. `CLS` token) that aggregates information from that key grow larger. This can result in an attention logit explosion and attention collapse as noted by Zhai et al. (2023). But this information can sometimes be a spurious correlation. In the attention operation, higher key norms increase the attention towards that key token and reduce the attention to other key tokens. The gradients through the attention operation are weighted by the attention probabilities (Katz & Wolf, 2025). Hence, tokens with lower key norms and thereby lower attention probabilities, also contribute less to parameter updates. Further, the parameter updates to the queries depend on a linear combination of the keys and vice versa. This indicates a cross-play where large key norms can further cause related query norms to increase, potentially leading to attention entropy collapse which contributes to training instabilities. We provide a more detailed theoretical analysis of this through the gradient updates corresponding to the queries and keys in A.1. Models initially learn features which are easy-to-learn, which can sometimes be spurious. If attention solely focuses on these features to solve the task, it is harder for the model to *unlearn* these spurious features and learn other useful features by attending to other tokens. We demonstrate this using a toy example in the section below, where we study the training of a simple Transformer model by introducing some spurious patterns in the training data.

In standard attention (Vaswani et al., 2017), the constant scaling factor $C = 1/\sqrt{D_H}$ was proposed to prevent large attention logit values, which were observed when $D_H$ was large. However, recent efforts to scale Vision Transformer models have shown that this formulation can still be unstable for large Transformer models due to arbitrarily growing attention logits (Dehghani et al., 2023). The proposed solution was to $\ell_2$-normalize both the queries and keys and scale each feature dimension in the queries and keys by learnable parameters (unique to each layer but shared across the heads), $\boldsymbol{C}_q, \boldsymbol{C}_k \in \mathbb{R}^{D_H}$. We refer to this as QKNorm-DS attention. Another related formulation, QKNorm-HS (Liu et al., 2022) instead scales each head with a learnable scalar $\boldsymbol{C} \in \mathbb{R}^H$. DS and HS denote dimension and head scaling, respectively. We list the limitations of these attention variants below:

1. **Standard attention** is known to have training instabilities arising from large attention logits (Zhai et al., 2023). Arbitrarily increasing key and query norms is one mechanism through which this happens.

2. **QKNorm attentions** (both QKNorm-HS and QKNorm-DS) enable stable training but scale all tokens in all heads by the same scaling factor which limits the expressivity of attention since all tokens are constrained to have the same sharpness.

A natural middle ground is to normalize either queries or keys. This would break the cross dependence between query (key) norms and key (query) gradient, with the potential of stabilizing the training. Neither of these options have however, to the best of our knowledge, been proposed in the literature. We will evaluate both options below, but what we propose is a new formulation of attention obtained by normalizing the keys while keeping the queries unnormalized. The intuition behind this choice is to allow each token to individually control the sharpness of its softmax distribution, and to prevent that large key norms "steal attention globally". We call this attention formulation, *Query-modulated Spherical Attention (QUEST)*, and compute attention as: $\boldsymbol{A} = \text{softmax}\left(\boldsymbol{Q}\bar{\boldsymbol{K}}^T\right)$, where $\bar{\boldsymbol{K}}$ denotes $\ell_2$-normalized keys. Note that we do not use any additional scaling, i.e. $C = 1$. This is an easily interpretable attention variant where the rank order of the attention distribution is purely defined by the vector alignment between queries and keys in the hyperspherical latent space (cosine similarity). The query norms allow each query to independently control the sharpness of its attention.

## 2.2 SPURIOUS ATTENTION ISSUES

We construct a simple toy example to demonstrate how standard attention can get stuck on spuriously correlated data patterns and find it difficult to learn the true and more consistent patterns in the data. Given a sequence of $N$ vectors $\boldsymbol{X} = [\boldsymbol{x}_1, ..., \boldsymbol{x}_N]$, where $\boldsymbol{x}_i \in \mathbb{R}^D$, the task is to retrieve information (an *answer*) from one of the vectors in $[\boldsymbol{x}_1, ..., \boldsymbol{x}_N]$.

The vectors consist of two parts: a real-valued vector $\boldsymbol{x}_i^k$ and a one-hot encoded vector $\boldsymbol{x}_i^v$. All the real-valued vectors are sampled from a certain distribution except the one at a random answer location $L$. A correctly learned model should learn to identify this "out of distribution" vector $\boldsymbol{x}_L^k$ and extract the answer $\boldsymbol{x}_L^v$ from its location. This is a robust signal that is always true but we introduce a biased signal that can also solve the task, but only for a subset of the samples ($\sim$50%).

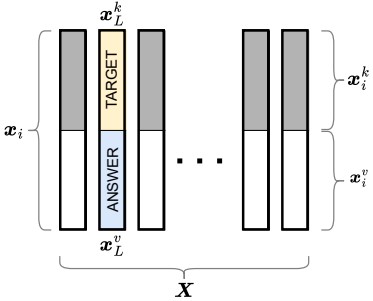

Specifically, *independently for each training sample*, let $L \sim$ $\mathrm{Int}(\mathcal{N}(\mu_l, \sigma_l))$ denote the location of the answer token and let $u \sim \mathrm{Bernoulli}(p = 0.5)$ denotes if the sample is biased or not. For all non-answer tokens we sample $\boldsymbol{x}_{i; i \neq L}^k \sim \mathcal{N}(0, \boldsymbol{I})$ (regardless of $u$), whereas for the answer token we sample

Figure 2: Illustration of toy example.

$\boldsymbol{x}_L^k \sim \mathcal{N}(0, \Sigma)$ if $u = 0$ (unbiased) or $\boldsymbol{x}_L^k \sim \mathcal{N}(\boldsymbol{b}, 0.1\boldsymbol{I})$ if $u = 1$ (biased). Here, the bias vector $\boldsymbol{b} \sim \mathcal{N}(0, \Sigma)$ is shared for all biased samples and $\Sigma \neq I$ (details in A.3). Note that we introduced an additional bias in the answer location (by sampling from a normal distribution with mean $\mu_l$ and standard deviation $\sigma_l$) to ensure that the inputs are still biased even after the addition of a positional embedding. The answer part of the vectors, $\boldsymbol{x}_i^v$, are all sampled as uniform one-hot vectors over $C$ classes, but it is the one-hot vector at location $L$ that is defined as the correct answer.

We consider a simple Transformer model $y = f(\boldsymbol{X}; \theta)$ with parameters $\theta$ which takes an input sequence $\boldsymbol{X}$ to produce a classification output $y$. We use a single Transformer layer (with learnable positional embeddings), which is sufficient to solve this task, and one head, which enables us to easily study the effect of different query and key norms. We use the [CLS] token features and use a linear layer to produce the class logits. More implementation details of the Transformer model is provided in the appendix A.3. We run this experiment with 5 different realizations of the data and 5 different weight initializations for the model weights. We train the model using the AdamW (Loshchilov & Hutter, 2018) optimizer for 50 epochs with a batch size of 32 and use learning rate values $\{0.0005, 0.001, 0.0025, 0.005, 0.0075, 0.01\}$ and weight decay values $\{0.0, 0.01, 0.02, 0.05, 0.1\}$.

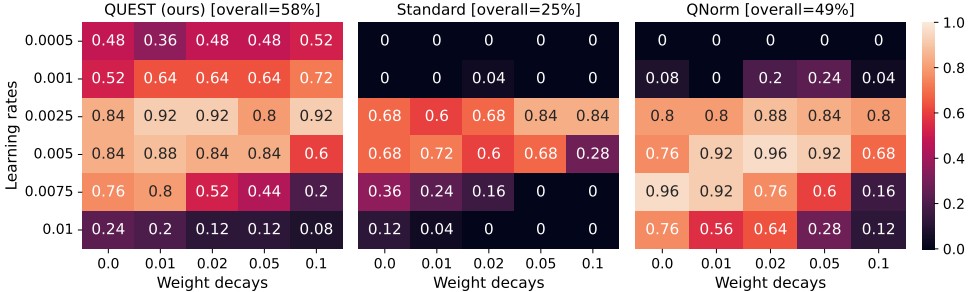

Figure 3: **Success rates of learning the correct solution to the toy example.** Models are trained with different hyperparameter combinations with 5 different weight initializations and 5 different realizations of the data. The QKNorm methods obtained $\sim$0% overall success rate and their results are available in Figure A1.

Based on the performance on the training and test sets, we can categorize the learned models as degenerate (both training and test accuracy are random chance), biased (training accuracy of $50 \sim 80\%$ but a test accuracy $\sim 20 - 40\%$) and correct (training and test accuracy $> 90\%$). The biased solution could learn a combination of the position and the vector bias to achieve a test accuracy $\sim 20 - 40\%$. We observed that degenerate solutions were common among QKNorm methods, which

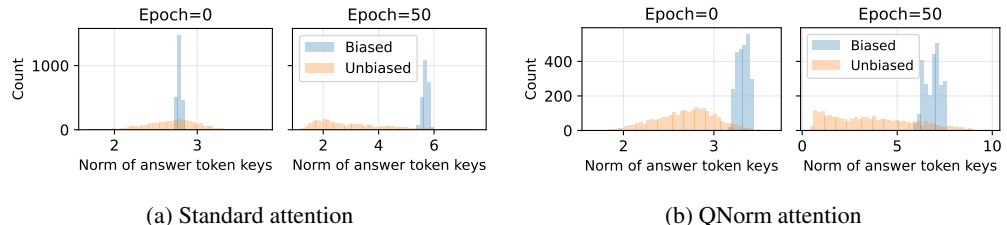

(a) Standard attention        (b) QNorm attention

Figure 4: **Norms of answer key tokens for biased and unbiased samples:** A common failure case for standard and QNorm attention involves the key norms of the biased answer token increasing as the training progresses. The model relies on looking up the bias vector to identify the answer.

is not surprising since they completely discard the information about the atypical distribution of the answer location contained in the norm of this vector. In general, the biased solution is easier to learn and only requires the model to look up the answer corresponding to the biased target vector $b$. Specifically, the weight matrix for the keys will align with the vector $b$ in the sense that the stretch (or amplification) factor is large in the direction of $b$, allowing the key norms to grow. This will concentrate attention globally on the answer location, but *only when the bias vector is present*. In standard attention, this is accelerated by the cross-play between query and key parameter updates. In QNorm, the queries are $\ell_2$-normalized, which helps in reducing this cross-play effect. Indeed, investigating the failed trainings of standard and QNorm attention, we found that the norms of the biased answer keys grew as the training progressed (see Figure 4). These models only learn to attend to the biased vector $b$ but fail to learn the correct solution. QUEST mitigates this effect by ensuring that individual tokens are unable to "steal attention" globally. In Figure 3, we show the success rates of learning the correct solution using different hyperparameter combinations and different attention formulations. We observe that the proposed QUEST attention displays a better overall success rate of 58% that also works well across a wider range of hyperparameter values.

## 3   RELATED WORK

In this work, we focus on improving the core attention mechanism and, hence, restrict our discussion to other works that explored this. We consider this work as an orthogonal contribution to other improvements in architecture design, training techniques and optimization. With the motivation of improving efficiency, a class of linear complexity Transformers without the softmax operation have been proposed (Wang et al., 2020; Choromanski et al., 2021; Kitaev et al., 2020; Katharopoulos et al., 2020; Han et al., 2023a) . However, this work explores softmax-based attention only. Probabilistic interpretations of attention have connected attention to Nadaraya-Watson regression with Gaussian isotropic kernels (Nguyen et al., 2022b; Han et al., 2023b), mixture models (Nguyen et al., 2022a), and asymmetric kernels (Chen et al., 2023). Elliptical attention (Nielsen et al., 2024) is a recent work that extends the Gaussian isotropic kernel interpretation of standard attention to hyper-ellipsoids using a Mahalanobis metric. QUEST attention uses keys in the hyperspherical latent space but we show that they are synergetic and can be combined as Elliptical QUEST. This uses an elliptical metric instead of cosine similarity between queries and keys (see A.5.2 for further discussion).

Prior works have studied the use of LayerNorm (Ba et al., 2016) in attention (Xiong et al., 2020) and its positioning (Wang et al., 2019). A main contributor to the training instability of Transformer models is the attention logit explosion (Zhai et al., 2023). This is directly related to the properties of queries and keys (Bao et al., 2024). Explicit $\ell_2$-normalization of the queries and keys, proposed as QKNorm (Dehghani et al., 2023; Liu et al., 2022), is the closest to our work but were mainly aimed at scaling Vision Transformers. Mongaras et al. (2025) used a similar formulation to QKNorm, but in the context of softmax-free attention. While this makes large models stable to train, we show that this limits the expressivity of attention, shown by worse performance on smaller models compared to standard attention. Our QUEST attention is applicable to Transformers in general, stable to train up to the scales that we have attempted and shows improved performance compared to QKNorm. Recent works on linear attention have also identified similar attention entropy collapse in the context of linear attention (Meng et al., 2025a;b). We leave the extension and study of our proposed QUEST attention to linear attention variants for future work.

Table 1: Ablation of different QK normalization methods for ImageNet classification (ViT-Tiny model trained using DeiT for 300 epochs) [$^\dagger$Liu et al. (2022), $^\ddagger$Dehghani et al. (2023)]. The Top-1 validation accuracies for ImageNet are the means over 3 independent training runs. The detailed statistics of these results are reported in Table A1.

| Attention | Scaling | IN-val Top-1 | IN-v2 Top-1 | IN-ReaL Top-1 | IN-C MCE ↓ | IN-A Top-1 | Top-5 |
|---|---|---|---|---|---|---|---|
| Standard | $1/\sqrt{D_h}$ | 72.6 | 60.6 | 80.4 | 55.7 | 8.2 | 32.9 |
| QUEST | - | **73.4** | **61.1** | **81.2** | **55.0** | **8.5** | **34.6** |
| QNorm | - | 72.7 | 60.8 | 80.6 | 55.3 | 8.2 | 34.5 |
| QKNorm-HS | $^\dagger\ \boldsymbol{C} \in \mathbb{R}^{L \times H}$ | 72.5 | 60.8 | 80.5 | 56.4 | 7.9 | 33.3 |
| QKNorm-DS | $^\ddagger\ \boldsymbol{C}_q, \boldsymbol{C}_k \in \mathbb{R}^{L \times D_h}$ | 71.6 | 59.7 | 79.6 | 57.4 | 7.2 | 31.5 |
| QKNorm | $\boldsymbol{C}_q, \boldsymbol{C}_k \in \mathbb{R}^{L \times H \times D_h}$ | 71.9 | 59.3 | 79.0 | 58.1 | 7.0 | 31.0 |

## 4 EXPERIMENTS

We primarily focus on applications using Vision Transformers but also conduct experiments on Transformers used in other domains such as language modeling, graph Transformers, general time series and pointclouds (pointcloud segmentation experiment is included in the appendix A.5.7). Our goal is to demonstrate broad applicability and effectiveness offered by a simple modification. We therefore study the impact of replacing standard attention with QUEST in popular attention-based architectures in different settings across multiple domains. Further, we provide ablation experiments comparing QUEST with QNorm and QKNorm attentions in the image classification, language modeling and time series experiments.

### 4.1 VISION APPLICATIONS

#### 4.1.1 CLASSIFICATION

**Ablation:** We conduct image classification experiments using the DeiT (Touvron et al., 2021a) training method. Firstly, we perform ablation experiments on different QK-normalization patterns in attention by training a ViT-Tiny model on ImageNet-1K dataset for 300 epochs and report the results in Table 1. In addition to evaluating on the ImageNet validation set, we also evaluate on the validation data from ImageNet-v2 (Recht et al., 2019), ImageNet-ReaL (Beyer et al., 2020), ImageNet-Adversarial (Hendrycks et al., 2021) and ImageNet-Corrupted (Hendrycks & Dietterich, 2019). We report validation accuracies on all datasets except for IN-C, which is evaluated using the mean corruption error (MCE) over 16 corruptions. We observe that the proposed QUEST attention performs clearly better than standard attention but also compared to alternative methods to normalize the queries and keys. While QKNorm-DS is shown to be stable for larger ViTs, we observe that it performs worse than standard attention in smaller models as it limits the expressivity of attention. Further, in Table A1, we show that these observations are statistically significant by repeating this experiment three times. A theoretical discussion about how QUEST is able to perform favorably, based on the gradients in the optimization process, is provided in appendix A.1. On this basis, we consider standard attention as the baseline in other experiments. If training is unstable with standard attention, then we use other QKNorm variants as the baseline.

**DeiT and DeiT-3:** Next, we consider experiments on larger sizes of Vision Transformers (Small, Base and Large). DeiT training with standard attention is unstable and divergent for ViT-Base and Large. In those cases, we use QKNorm-DS attention as the baseline. The shorter 100 epoch trainings on ViT-Base and ViT-Large models show that QUEST attention is stable to train in contrast to standard attention and also achieves better performance compared to QKNorm attention (see Table A6 for an additional experiment on a 2B parameter ViT which shows consistent results). We empirically analyze the progression of maximum attention logits and the corresponding query/key norms in Figure A6. Although only the key tokens were normalized in QUEST, we observe that this also stabilizes the query norms and consequently the attention logits, thus enabling stable training. DeiT-3 proposed an improved and stable training recipe for larger ViTs by using better augmentation strategies and techniques such as stochastic depth (Huang et al., 2016) and LayerScale (Touvron et al., 2021b). We also evaluate QUEST attention using DeiT-3 training for the Base, Large and Huge ViTs, that

Table 2: ImageNet Classification using DeiT and DeiT-3 training recipes [‡Dehghani et al. (2023)]. For longer trainings with larger ViT models (denoted as †), the batch size and input image resolution in the first phase of training are changed compared to Touvron et al. (2022) in order to train on a limited compute infrastructure, see A.5.1 for details.

| Model | Attention | Epochs | IN-val Top-1 | IN-v2 Top-1 | IN-ReaL Top-1 | IN-C MCE ↓ | IN-A Top-1 | IN-A Top-5 |
|---|---|---|---|---|---|---|---|---|
| DeiT-1 Touvron et al. (2021a) trainings | | | | | | | | |
| ViT-S/16 | Standard | 200 | 79.6 | 68.4 | 85.8 | 44.8 | 18.2 | 50.7 |
| ViT-S/16 | QUEST | 200 | **80.2** | **68.9** | **86.2** | **43.2** | **20.4** | **53.5** |
| ViT-B/16 | Standard | 100 | | | –training crashed– | | | |
| ViT-B/16 | QKNorm-DS‡ | 100 | 79.0 | 67.5 | 84.9 | 44.4 | 17.7 | 50.0 |
| ViT-B/16 | QUEST | 100 | **79.7** | **68.7** | **85.7** | **42.9** | **19.2** | **51.9** |
| ViT-L/16 | Standard | 100 | | | –training crashed– | | | |
| ViT-L/16 | QKNorm-DS‡ | 100 | 72.5 | 58.5 | 78.2 | 54.4 | 8.9 | 31.2 |
| ViT-L/16 | QUEST | 100 | **74.9** | **61.4** | **80.6** | **50.3** | **11.1** | **35.6** |
| DeiT-3 Touvron et al. (2022) trainings | | | | | | | | |
| ViT-B/16 | Standard | 400 + 20 | 82.7 | 72.4 | 88.0 | 36.5 | 34.8 | 67.7 |
| ViT-B/16 | QUEST | 400 + 20 | **83.2** | **73.3** | **88.2** | **35.7** | **37.6** | **69.9** |
| ViT-L/16† | Standard | 400 + 20 | 83.9 | 74.0 | 88.7 | 32.5 | 44.1 | 75.3 |
| ViT-L/16† | QUEST | 400 + 20 | **84.1** | **74.3** | **88.9** | **32.3** | **44.5** | **76.0** |
| ViT-H/14† | Standard | 400 + 20 | 83.2 | 73.5 | 88.6 | 34.1 | 45.9 | 77.7 |
| ViT-H/14† | QUEST | 400 + 20 | **83.4** | **74.0** | **88.7** | **33.7** | **46.2** | **78.0** |

were training for a larger number of epochs until convergence. The results are reported in Table 2. For all the considered model sizes, we find that QUEST consistently achieves better performance. We observe larger performance improvements in IN-C and IN-A evaluations, showing that QUEST attention produces more robust models. We further explore this through experiments on adversarial robustness and explainability in 4.1.2 and 4.1.4. We show additional experiments in A.5.1 to evaluate the sensitivity of these results to changes in the training data and hyperparameters like learning rate. We found the performance improvement to be consistent when training with different limited data subsets. The models with QUEST attention are stable to train at different learning rates.

**Additional experiments:** We experiment with an alternate Transformer architecture, CrossViT (Chen et al., 2021), to demonstrate that QUEST attention is compatible with cross-attention (see A.5.1). Since it is commonplace to use self-supervised models, we also evaluate QUEST attention for pre-training in A.5.9, and find that it improves over standard attention in downstream performance.

### 4.1.2 ROBUSTNESS

In addition to improved robustness to corruptions and adversarially curated images observed above, we evaluate the robustness of QUEST attention to adversarial attacks. For adversarial attacks under image perturbations, we adopt the experimental setup of Nielsen et al. (2024). We report the validation performance under the following adversarial attacks: fast gradient sign method (FGSM) (Goodfellow et al., 2015), PGD attack based on projected gradient descent (Madry et al., 2018), SPSA attack based on simultaneous perturbation stochastic approximation (Uesato et al., 2018) and Auto attack (Croce & Hein, 2020). The Auto attack is an ensemble of auto PGD-Cross Entropy, auto PGD-targeted , fast adaptive boundary-targeted and Square attacks. The complete experimental details are provided in A.5.2. We report the validation accuracies and NLL, after adversarial perturbations for the ViT-Tiny model in Table 3 and 4 (see Table A5 for similar results for larger ViT models). Firstly, we observe that QUEST is more robust than standard attention across all adversarial attacks. QUEST attention focuses more evenly on relevant object regions whereas standard attention concentrates on only a few object parts or object instances (see further discussion below in 4.1.4). Elliptical attention (Nielsen et al., 2024) is a SOTA model for robustness but this is achieved at the expense of classification accuracy (71.53% vs 72.50%). We show that QUEST can be orthogonally added to Elliptical attention to further improve robustness, while also achieving better classification accuracy on clean data.

Table 3: Robustness to adversarial attacks using ViT-Ti/16 model trained using DeiT

| Attention | Clean Data | | | FGSM | | | PGD | | |
|-----------|-------|-------|-------|-------|-------|-------|-------|-------|-------|
| | Top-1 | Top-5 | NLL | Top-1 | Top-5 | NLL | Top-1 | Top-5 | NLL |
| Standard | 72.50 | 91.45 | 1.190 | 54.23 | 85.28 | 1.827 | 43.65 | 78.18 | 2.503 |
| QUEST | **73.33** | **91.91** | **1.160** | **56.90** | **86.63** | **1.745** | **45.26** | **79.33** | **2.448** |
| Elliptical | 71.53 | 90.70 | 1.254 | 55.96 | 85.53 | 1.746 | 46.30 | 80.05 | 2.231 |
| Elliptical-QUEST | **72.48** | **91.20** | **1.214** | **56.39** | **85.94** | **1.741** | **47.25** | **80.61** | **2.211** |

Table 4: Robustness to adversarial attacks using ViT-Ti/16 model trained using DeiT

| Attention | SPSA | | | Auto | | |
|-----------|-------|-------|-------|-------|-------|-------|
| | Top-1 | Top-5 | NLL | Top-1 | Top-5 | NLL |
| Standard | 47.17 | 83.95 | 2.027 | 26.57 | 67.60 | 3.230 |
| QUEST | **50.70** | **85.49** | **1.904** | **27.29** | **67.98** | **3.200** |
| Elliptical | 57.96 | 86.92 | 1.654 | 27.35 | 67.28 | 3.014 |
| Elliptical-QUEST | **59.15** | **87.44** | **1.613** | **28.54** | **68.10** | **2.965** |

### 4.1.3 SEGMENTATION

We evaluate image segmentation using the Segmenter approach (Strudel et al., 2021) using a Mask Transformer decoder, where we initialize the backbone ViT model with the DeiT weights from above and finetune the entire model (encoder and decoder) for semantic segmentation on the ADE20K dataset (Zhou et al., 2019; 2017). We also evaluate the segmentation models for robustness under 16 different types of image corruptions, following the experimental setup of Zhou et al. (2022). The segmentation results are reported in Table 5. Based on the commonly used mIoU metric (mean Intersection over Union), QUEST attention performs better than standard attention and displays better robustness to corruptions.

Table 5: ADE20K Image Segmentation

| Model | Attention | Clean data mIoU | Corrupted data mIoU |
|-------|-----------|-----------------|---------------------|
| ViT-Ti/16 | Standard | 37.34 | 32.19 |
| ViT-Ti/16 | QUEST | **38.87** | **33.55** |
| ViT-S/16 | Standard | 43.43 | 38.45 |
| ViT-S/16 | QUEST | **44.13** | **39.19** |

### 4.1.4 EXPLAINABILITY

Attention-based models are also beneficial from a model explainability standpoint. AG-CAM (Leem & Seo, 2024) is a recent explainability method that combines attention maps and gradient information to produce class activation maps (CAMs) for image classifiers. Following the same evaluation protocol, we apply a probability threshold of 0.5 to the CAMs and evaluate pixel accuracy, mIoU and DICE score by comparing with ground truth localization labels in ImageNet. We consider the ViT-B model trained using the DeiT-3 training recipe and report the results for QUEST and standard attention in Table 6. In Figure 1 and 5, we show how a model trained with QUEST produces a better CAM for different classes. Specifically, standard attention concentrates on few object parts or instances whereas QUEST attends to them more evenly (see A.5.3 for additional qualitative examples). By not relying on only a few aspects of an object, QUEST can perform more robustly as observed in 4.1.2.

Table 6: Model explainability using AG-CAM

| Backbone | Attention | Method | Epochs | Pixel accuracy (%) ↑ | mIoU ↑ | DICE Score ↑ |
|----------|-----------|--------|--------|----------------------|--------|--------------|
| ViT-B/16 | Standard | DeiT-3 | 400 + 20 | 65.22 | 35.53 | 0.4870 |
| ViT-B/16 | QUEST | DeiT-3 | 400 + 20 | **70.76** | **53.54** | **0.6597** |

## 4.2 TIME-SERIES CLASSIFICATION

We conduct general sequence classification experiments using the UEA multivariate time series classification suite (Bagnall et al., 2018). We train Transformer models using the experimental setup in Wang et al. (2024) (see results in Table 7; more details in A.5.6). With the exception of 3 datasets, QUEST attention performs better or the same as standard attention. Through an ablation experiment in Table A8, we show that QUEST also outperforms QNorm and QKNorm based on the overall average performance. The overall average accuracy of QUEST attention surpassed the SOTA (73.17%) achieved by Crossformer Zhang & Yan (2023).

## 4.3 GRAPH TRANSFORMERS

Recently, several Transformer-based models have been proposed for tasks involving graph-structured data. We adopt the experimental setup of GraphGPS (Rampášek et al., 2022) to evaluate on standard GNN benchmarks from Dwivedi et al. (2023) and long-range graph benchmarks (LRGB) from Dwivedi et al. (2022). We use the same optimal hyperparameter setups for each dataset as in GraphGPS (see A.5.8 for detailed experimental setups for each dataset) and evaluate QUEST attention as a drop-in replacement for standard attention used in those graph Transformers. The results for the standard and LRGB benchmarks are reported in Tables 8 and 9, respectively. For the standard benchmarks, standard attention and QUEST perform on par for all datasets (within significance range). On the long-range benchmark, we see significant improvements for COCO-SP, Peptides-func and PCQM-Contact, whereas we perform on par with standard attention for PascalVOC-SP and Peptides-struct.

Table 8: GNN Benchmarks using GraphGPS (mean $\pm$ s.d over 10 runs)

| Attention | ZINC | CIFAR10 | PATTERN | CLUSTER |
|---|---|---|---|---|
| | MAE $\downarrow$ | Accuracy $\uparrow$ | Accuracy $\uparrow$ | Accuracy $\uparrow$ |
| Standard | $0.070 \pm 0.004$ | $72.298 \pm 0.356$ | $86.685 \pm 0.059$ | $\mathbf{78.016 \pm 0.180}$ |
| QUEST | $\mathbf{0.069 \pm 0.002}$ | $\mathbf{72.843 \pm 0.526}$ | $\mathbf{86.760 \pm 0.046}$ | $77.894 \pm 0.205$ |

Table 9: Long Range Graph Benchmarks using GraphGPS (mean $\pm$ s.d over 4 runs)

| Attention | PascalVOC-SP | COCO-SP | Peptides-func | Peptides-struct | PCQM-Contact |
|---|---|---|---|---|---|
| | F1 score $\uparrow$ | F1 score $\uparrow$ | AP $\uparrow$ | MAE $\downarrow$ | MRR $\uparrow$ |
| Standard | $\mathbf{0.375 \pm 0.011}$ | $0.341 \pm 0.004$ | $0.654 \pm 0.004$ | $\mathbf{0.250 \pm 0.001}$ | $0.334 \pm 0.001$ |
| QUEST | $0.373 \pm 0.003$ | $\mathbf{0.349 \pm 0.004}$ | $\mathbf{0.662 \pm 0.004}$ | $0.251 \pm 0.002$ | $\mathbf{0.346 \pm 0.001}$ |

Figure 5: **Class activation maps for Elephant and Zebra.** Model with QUEST attention shows better coverage of the different instances of the animals than standard attention.

Table 7: UEA Multivariate Time Series Classification using Transformers

| Dataset | Standard | QUEST |
|---|---|---|
| EthanolConcentration | 29.28 | **30.42** |
| FaceDetection | 65.24 | **65.83** |
| Handwriting | 42.00 | **49.18** |
| Heartbeat | 77.56 | **78.54** |
| JapaneseVowels | **98.38** | **98.38** |
| PEMS-SF | **83.82** | 80.92 |
| SelfRegulationSCP1 | **88.05** | **88.05** |
| SelfRegulationSCP2 | **58.89** | 58.33 |
| SpokenArabicDigits | 98.86 | **99.41** |
| UWaveGestureLibrary | **86.88** | 86.25 |
| Average | 72.90 | **73.53** |

## 4.4 LANGUAGE MODELING

We conduct language modeling experiments using the WikiText-103 dataset (Merity et al., 2017) and use the experimental setup from Nielsen et al. (2024) and (Schlag et al., 2021) for the Transformer model. We also consider a larger model, the Transformer-XL (Dai et al., 2019) and evaluate it using their experimental setup. Additionally, we evaluate the robustness of these language models using the Word Swap Attack from Nielsen et al. (2024) that replaces random words with the "AAA" token with specified rates. The perplexity (PPL) metrics for these models using clean and contaminated data (different attack rates) are reported in Table 10. We observe that QUEST attention generally produces marginally better performance on both clean and contaminated data. When comparing QUEST with QNorm and QKNorm on the Transformer-Medium model, we observe that QUEST performs favorably.

Table 10: Language Modeling on WikiText-103

| Method | Model size (Parameters) | Attention | Clean Data PPL ↓ | | Contaminated Data PPL (Corruption %) ↓ | | |
|---|---|---|---|---|---|---|---|
| | | | Val | Test | Test (1.5%) | Test (2.5%) | Test (5.0%) |
| Transformer | Small (44M) | Standard | 33.073 | 34.076 | 41.566 | 46.380 | **60.944** |
| Transformer | Small (44M) | QUEST | **32.966** | **33.966** | **41.522** | **46.299** | 61.062 |
| Transformer | Medium (90M) | Standard | 27.441 | 28.851 | 36.234 | 40.866 | 55.012 |
| Transformer | Medium (90M) | QNorm | 27.233 | 28.688 | 36.262 | 40.853 | 55.451 |
| Transformer | Medium (90M) | QKNorm-DS | 27.376 | 28.624 | 36.018 | 40.715 | 54.899 |
| Transformer | Medium (90M) | QUEST | **26.980** | **28.478** | **35.849** | **40.499** | **54.531** |
| Transformer-XL | Base (151M) | Standard | 22.650 | 23.592 | 29.627 | 33.386 | 44.168 |
| Transformer-XL | Base (151M) | QUEST | **22.436** | **23.320** | **29.339** | **33.008** | **43.511** |

## 5 CONCLUSION

The instabilities in training Transformers are well known and occur when attention collapses due to arbitrarily increasing query and key norms. We demonstrate how spuriously correlated features in certain tokens can contribute to such behavior. Unlike prior works which argued that such issues only occurred in larger models, we showed that they can also limit small models, resulting in reduced performance even if the training does not diverge. We propose a simple drop-in replacement called QUEST attention that considers a hyperspherical latent space for attention while still allowing individual tokens to flexibly and independently control the sharpness of their attention distributions. Through extensive experiments on several domains, we demonstrate broad applicability. QUEST attention trains more robustly and produces models that typically perform better and that are more robust to corruptions and adversarial attacks than standard attention and its variants like QKNorm. We also show that QUEST attention improves the robustness of the trained models and can orthogonally improve SOTA methods like Elliptical attention. While we have shown consistent improvements when using QUEST with Transformers in multiple domains, we have primarily focused on vision applications and leave more in-depth evaluation on other domains and incorporation of QUEST into domain-specific SOTA architectures for future work. Given the effectiveness of operating in the hyperspherical latent space, exploring more geometrically aligned operations and optimization methods, e.g. Riemannian gradient descent (Kasai et al., 2019), is another promising avenue for future work.

ACKNOWLEDGMENTS

This research is financially supported by the Swedish Research Council (project no: 2024-05011), the Wallenberg AI, Autonomous Systems and Software Program (WASP) funded by the Knut and Alice Wallenberg Foundation, and the Excellence Center at Linköping–Lund in Information Technology (ELLIIT). The computations were enabled by the Berzelius resource provided by the Knut and Alice Wallenberg Foundation at the National Supercomputer Centre.

## REPRODUCIBILITY STATEMENT

We include an algorithmic implementation of QUEST attention and standard attention in Algorithm A1 to clearly illustrate our proposed modification. In all our experiments, we use standard evaluation protocol and use publicly available code repositories. We introduced QUEST attention as a simple drop-in replacement for standard attention in the different experiments that we conducted. We provide details in the Appendix (see A.5) for all the code repositories used and clarify any changes to hyperparameter configurations.

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

# A APPENDIX

## A.1 THEORETICAL ANALYSIS OF ATTENTION GRADIENT UPDATES

In the context of Transformer gradient updates, Katz & Wolf (2025) defined a reverse attention term, $\boldsymbol{R}$ as follows:

$$\tilde{\boldsymbol{E}} = \boldsymbol{\Delta} \boldsymbol{W}_o^T \boldsymbol{V}^T \in \mathbb{R}^{N \times N}$$

$$\boldsymbol{R} = \boldsymbol{A} \odot \left( \tilde{\boldsymbol{E}}^T - \mathrm{diag}(\boldsymbol{A}\tilde{\boldsymbol{E}}^T) \right)^T \sqrt{\frac{1}{D_H}} \in \mathbb{R}^{N \times N}$$

where $\boldsymbol{A}$ is the attention computed in the forward pass, $\boldsymbol{W}_o$ is the output projection weight and $\boldsymbol{\Delta} \in \mathbb{R}^{N \times D}$ is the Vector Jacobian Products (VJP) of $\boldsymbol{W}_o$. Note that when attention, $\boldsymbol{A}$ concentrates on specific tokens, the reverse attention $\boldsymbol{R}$ also concentrates on those tokens (contributions from other tokens approach 0). This term is then used to derive the VJPs for the query and key gradient updates (for token $j$) as:

$$\boldsymbol{\delta}_q^j = \boldsymbol{R}_j \boldsymbol{K} \in \mathbb{R}^{D_H}$$

$$\boldsymbol{\delta}_k^j = \boldsymbol{R}_j^T \boldsymbol{Q} \in \mathbb{R}^{D_H}$$

The VJPs of the queries and keys are linear combinations of the keys and queries respectively. These VJPs are dominated by the tokens where attention concentrated and the contributions from the other tokens are diminished. In the attention operation, higher key norms increase the attention towards that key token and reduce the attention to other key tokens. Hence, tokens with lower key norms and thereby lower attention probabilities, also contribute less to parameter updates. When key norms of these tokens grow, this can consequently cause the query norms attending to that token to grow as well. This cross-play causes query and key norms to feed off each other and continue growing, resulting in an attention logit collapse.

Based on the above observation, $\ell_2$-normalizing at least one of them can mitigate this effect. This can also be empirically observed in Figure A6 in the paper - note that the query norms and max logits in QUEST are stable and do not increase as in the case of standard attention. This provides an explanation as to why QUEST, QNorm and QKNorm are all able to produce stable trainings. Note that, among these, QKNorm, is the only option that has previously been proposed in the literature (Dehghani et al., 2023; Liu et al., 2022) as a stabilizing modification of attention, to the best of our knowledge. QNorm and QUEST provide additional flexibility in the attention mechanism and hence perform better compared to QKNorm. QNorm, though stable, can still allow key norms of specific tokens to grow and hence, "steal attention" from other tokens. On the other hand, the query norms in QUEST can only influence the sharpness of attention and not which token should be attended to. We believe that this enables QUEST to learn better attention distributions as shown in the class-activation maps (see Figures 1,A7 and A8) and perform robust under corruptions and adversarial attacks.

## A.2 QUEST ATTENTION IMPLEMENTATION

The implementation of both standard attention and QUEST attention is illustrated in Algorithm A1. The key modification in QUEST attention is to $\ell_2$-normalize the keys in lines 16-17. In any method that currently uses standard attention, QUEST attention can be used as a drop-in replacement. For other variants of attention that still use a similar scaled dot-product formulation, a QUEST attention variant can be obtained by normalizing the keys.

## A.3 IMPLEMENTATION DETAILS OF THE TOY EXAMPLE

In this section, we provide additional details about the construction of the toy example and the Transformer model used in our experiments.

**Toy example construction:** In the toy example, the real-valued vectors at the answer location $\boldsymbol{x}_L^k$ are sampled differently depending on whether the sample is biased or not (as per the Binomial random variable $u$). For the unbiased case, we define $\Sigma = \boldsymbol{S}\boldsymbol{S}^T$, where all elements of $\boldsymbol{S}$ are sampled from a standard normal distribution. For the position bias, we sampled the answer positions from a normal distribution, $\mathcal{N}(\mu_l, \sigma_l)$ and converted them into integer indices. We used $\mu_l = 10$ and $\sigma_l = 2$. The most frequently sampled position occurs approximately 20% of the time in the data. The real-valued

---

**Algorithm A1** Computation of standard and QUEST attention

---

1: **Input:**
2: Tensor $Q_h \in \mathbb{R}^{N \times D_H}$                                                 $\triangleright$ Queries for $N$ tokens in head $h$
3: Tensor $K_h \in \mathbb{R}^{N \times D_H}$                                                  $\triangleright$ Keys for $N$ tokens in head $h$
4: Tensor $V_h \in \mathbb{R}^{N \times D_H}$                                              $\triangleright$ Values for $N$ tokens in head $h$
5: integer $D_H \in \mathbb{N}$                                                       $\triangleright$ Head dimension

6: **function** STANDARDATTENTION($Q_h, K_h, V_h$)
7:      $C \leftarrow \frac{1}{\sqrt{D_H}}$                                             $\triangleright$ constant scaling factor
8:      `attention_logits` $\leftarrow C \times Q_h \times K_h^T$
9:      `attention` $\leftarrow$ softmax(`attention_logits`)      $\triangleright$ softmax along the keys dimension
10:     `attention` $\leftarrow$ dropout(`attention`)                 $\triangleright$ attention dropout
11:     `output` $\leftarrow$ `attention` $\times V_h$
12:     `output` $\leftarrow$ `out_projection`(`output`)           $\triangleright$ linear output projection
13:     `output` $\leftarrow$ dropout(`output`)                     $\triangleright$ output dropout
14:     **return** `output`

15: **function** QUESTATTENTION($Q_h, K_h, V_h$)
16:     `NormK_factor` $\leftarrow$ diag $\left( \frac{1}{\|K_{h,1}\|}, ..., \frac{1}{\|K_{h,N}\|} \right)$ $\triangleright$ Calculate normalization factor of $N$ keys
17:     $\bar{K}_h \leftarrow$ `NormK_factor` $\times K_h$                         $\triangleright$ Compute normalized keys
18:     `attention_logits` $\leftarrow Q_h \times \bar{K}_h^T$
19:     `attention` $\leftarrow$ softmax(`attention_logits`)      $\triangleright$ softmax along the keys dimension
20:     `attention` $\leftarrow$ dropout(`attention`)                 $\triangleright$ attention dropout
21:     `output` $\leftarrow$ `attention` $\times V_h$
22:     `output` $\leftarrow$ `out_projection`(`output`)           $\triangleright$ linear output projection
23:     `output` $\leftarrow$ dropout(`output`)                     $\triangleright$ output dropout
24:     **return** `output`

---

vectors and the one-hot encoded vectors are both of 10 dimensions. As a result the task is a 10-way classification problem.

**Transformer model setup:** We use a one-layer Transformer model for the experiments involving the toy example as it is sufficient to solve the task. We use only one head to enable easier interpretation of the norms of keys and queries belonging to specific positions such as [CLS] and answer location. The Transformer model setup is illustrated in Algorithm A2 and follows the standard implementations that are commonly used. The embedding dimensions of the Transformer model is the same as the input data dimensions ($= 20$). The same number of dimensions is also used in the MLP hidden layer in the Transformer. For the different attentions, the only change is to use different attention functions (see examples for standard and QUEST attention in Algorithm A1).

---

**Algorithm A2** Transformer model setup in the toy example

---

1: **Input:**
2: Tensor $\boldsymbol{X} \in \mathbb{R}^{N \times D}$                                   ▷ Input data samples
3: Tensor $\boldsymbol{P} \in \mathbb{R}^{(N+1) \times D}$           ▷ Learnable positional embeddings for the $N$ positions
4: Tensor $\boldsymbol{X}_{\mathrm{CLS}} \in \mathbb{R}^{1 \times D}$                          ▷ Learnable CLS token

5: **function** TRANSFORMER($\boldsymbol{X}, \boldsymbol{X}_{\mathrm{CLS}}, \boldsymbol{P}$)
6:      $\boldsymbol{X} \leftarrow \texttt{PrependCLSToken}(\boldsymbol{X}_{\mathrm{CLS}}, \boldsymbol{X})$     ▷ Prepend a learnable CLS token to the data
7:      $\boldsymbol{X} = \boldsymbol{X} + \boldsymbol{P}$                                     ▷ Add positional embeddings
8:      $\boldsymbol{Y} \leftarrow \texttt{LayerNorm1}(\boldsymbol{X})$
9:      $\boldsymbol{Q} = \boldsymbol{X}\boldsymbol{W}_Q^T$                                        ▷ query projection
10:      $\boldsymbol{K} = \boldsymbol{X}\boldsymbol{W}_K^T$                                      ▷ key projection
11:      $\boldsymbol{V} = \boldsymbol{X}\boldsymbol{W}_V^T$                                     ▷ value projection
12:      $\boldsymbol{Y} \leftarrow \boldsymbol{Y} + \texttt{Attention}(\boldsymbol{Q}, \boldsymbol{K}, \boldsymbol{V})$
13:      $\boldsymbol{Y} \leftarrow \boldsymbol{X} + \texttt{LayerNorm2}(\boldsymbol{Y})$
14:      $\boldsymbol{Y} \leftarrow \boldsymbol{X} + \texttt{MLP}(\boldsymbol{Y})$
15:      $\texttt{output} \leftarrow \texttt{classifier}(\boldsymbol{Y}_{\mathrm{CLS}})$     ▷ Apply linear classifier to the CLS token features
16:      **return** $\texttt{output}$

---

### A.4 EXTENDED ANALYSIS OF THE TOY EXAMPLE

We provide some additional results and analysis using the toy example in this section. In the main paper, we only showed the test success rates for the 3 attention formulations that performed reasonably well (QUEST, Standard and QNorm attentions). In Figure A1 we show the success rates for all the attention formulations. We find that the QKNorm attentions largely fail achieving $\sim$0% overall success rate. We attribute this to the fact that they discard useful information by normalizing the queries and keys. The input vector norms provide an initial clue to solving the task.

The distributions for the training and test accuracies for the toy example are shown in Figures A2 and A3 respectively. We can clearly identify the biased and correct solutions based on their training and test accuracies. A biased solution achieves a training accuracy of $\sim$50-80% and a test accuracy of $\sim$20-40%. A correct solution achieves a test accuracy greater than 90%. Degenerate solutions are characterized by a random chance test accuracy 10%. In Figure A4 and A5, we show the comparison of norms of unbiased answer token keys and non-answer token keys. For both standard and QNorm attention, the models do not seem to distinguish between the unbiased answer tokens and non-answer tokens in terms of norms of their keys. On the other hand, they assign a much higher norm to the biased answer token keys, as shown in Figure 4 from the main paper.

### A.5 ADDITIONAL EXPERIMENTAL DETAILS

#### A.5.1 IMAGE CLASSIFICATION

**Investigating divergent DeiT trainings:** We found the DeiT training to be unstable for the ViT-Base model when trained on 4 NVIDIA A100 40GB GPUs using an overall batch size of 1024 as proposed in Touvron et al. (2021a). We use PyTorch 2.1 for this experiment. In Figure A6, we provide an

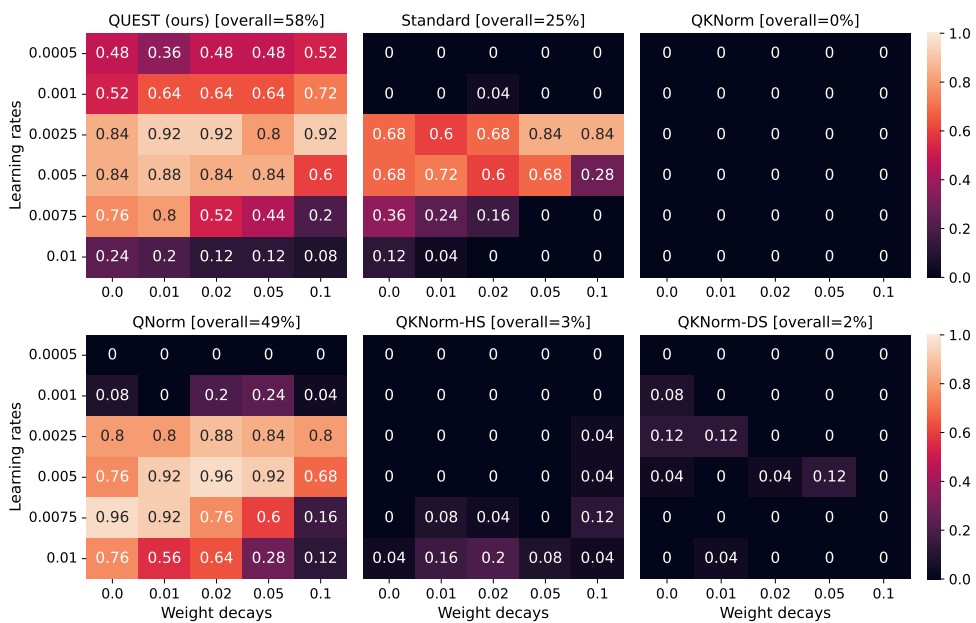

Figure A1: Success rates of learning the correct solution to toy example. Models are trained with different hyperparameter combinations with 5 different weight initializations and 5 different realizations of the data.

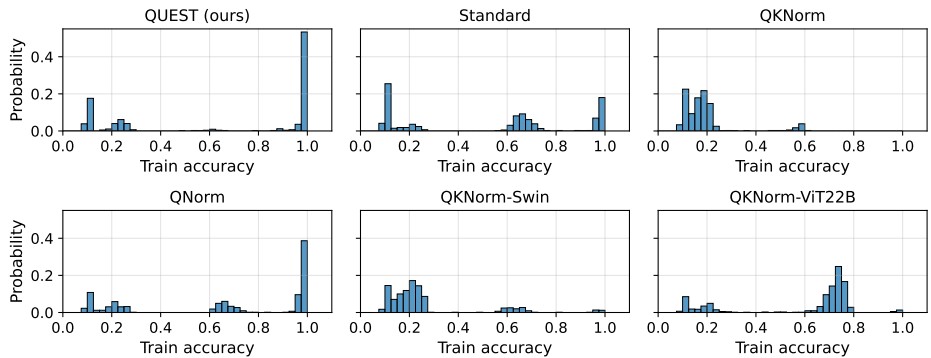

Figure A2: Distribution of training accuracies at the end of the training for the toy example.

analysis of the maximum logits and its associated query and key norms for a ViT-Base model trained using DeiT.

**Repeating DeiT ablation experiments:** For the ablation experiment shown in section 4.1.1 and Table 1 of the paper, we conduct three independent runs and show the aggregate results (mean and standard deviation) in Table A1. On running a 2-group t-test to check the statistical significance, we obtained p-values lower than 0.01 compared to the second best for all metrics in Table A1.

**Training with limited training data:** In Table A2, we show the results for training DeiT-Tiny models using limited subsets of the ImageNet training data. The data subsets are uniformly sampled and the reported results are aggregated over 3 such random subsets. All models are trained for 300 epochs with the same training setup as a standard DeiT model trained on the full ImageNet training dataset. We observe that QUEST consistently performs better than standard attention by 1.0-1.5%.

**Training with different learning rates:** In Table A3, we evaluate DeiT training using a ViT-Small model using different learning rates. We find standard attention to be unstable at larger learning rates.

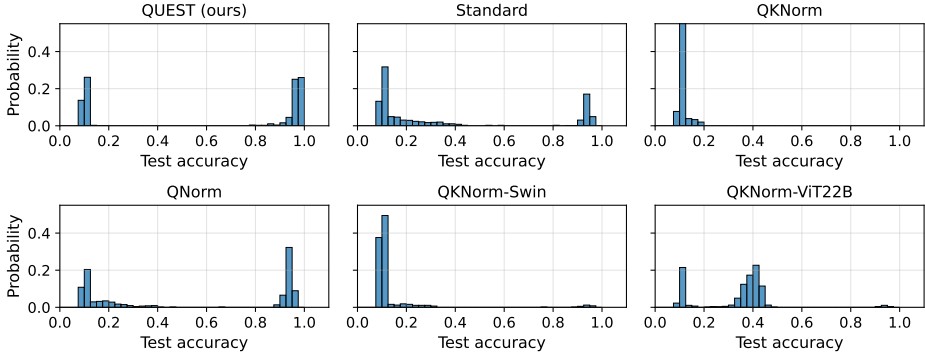

Figure A3: Distribution of test accuracies at the end of the training for the toy example.

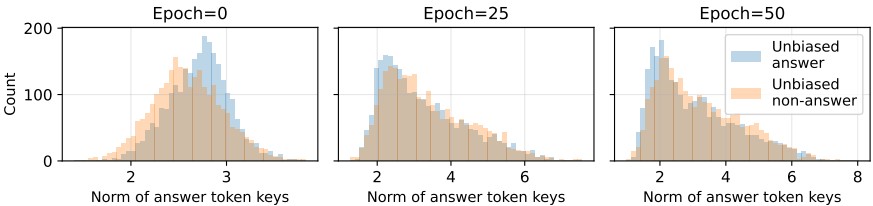

Figure A4: Norm of unbiased answer token keys and non-answer token keys in standard attention. In terms of the key norms, we do not observe any distinction between unbiased answer tokens and non-answer tokens.

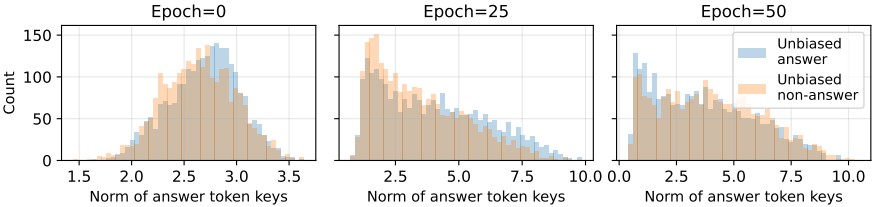

Figure A5: Norm of unbiased answer token keys and non-answer token keys in QNorm attention. In terms of the key norms, we do not observe any distinction between unbiased answer tokens and non-answer tokens.

QUEST attention produces stable training at different learning rates and consistently displays better performance.

**Training with large-scale ViT models:** In Table A6, we train large-scale ViT models, namely, ViT-Huge (600 Million parameters) and ViT-2B (2.4 Billion parameters) for a smaller number of epochs of 100 (+20 finetuning epochs) and 10 respectively using the stable DeiT-3 training recipe. We reduce the global batch sizes to 1024 for the ViT-Huge model and to 64 for the ViT-2B model, to fit on a single NVIDIA A100 node. For the ViT-Huge model, we use an input image resolution of $154 \times 154$ as in Touvron et al. (2022) but for the ViT-2B model, we use a reduced input image resolution of $96 \times 96$. Similarly, for the longer experiments (DeiT-3 trainings for 400+20 epochs) shown in Table 2 for ViT-Large and ViT-Huge, we made the following modifications compared to the standard training recipe:

- **ViT-L/16:** Batch size reduced from 2048 to 1024.

- **ViT-H/14:** Batch size reduced from 2048 to 1024, input image resolution in the first training phase reduced from $154 \times 154$ to $126 \times 126$.

**CrossViT, an architecture using cross-attention:** CrossViT (Chen et al., 2021) is an extension to ViT that uses two branches of Transformer layers using different patch size. This is followed by additional Transformer layers which use cross-attention instead of self-attention. We train the models using a 8 NVIDIA A100 40GB GPUs. We use the same experimental configuration as in Chen et al. (2021) for the different models and adapt the batch sizes to fit our 8 GPU setup. We consider the CrossViT-9-Dagger, CrossViT-Small and CrossViT-18-Dagger models. We use a batch size of 2048, 1024 and 1024 for the CrossViT-9-Dagger, CrossViT-Small and CrossViT-18-Dagger models respectively. We report the results using standard and QUEST attention in Table A4. We found the CrossViT-18-Dagger model to be unstable with standard attention and the training crashed

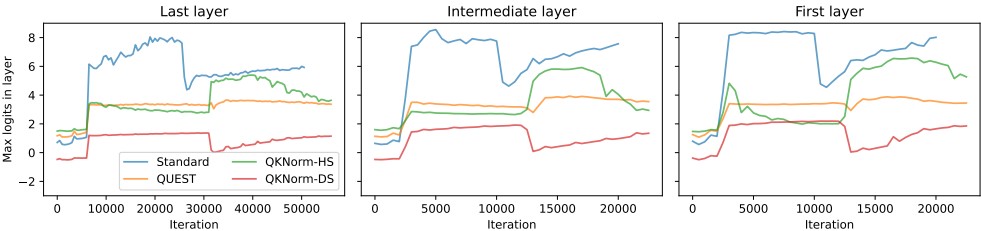

(a) Progression of maximum logits in the first, intermediate and last layers of a ViT-Base model trained using DeiT and using different attention formulations. We observe that the max logits are highly stable in different layers using QUEST attention. The max logits rapidly increase for standard attention before the training crashes around 20000 iterations.

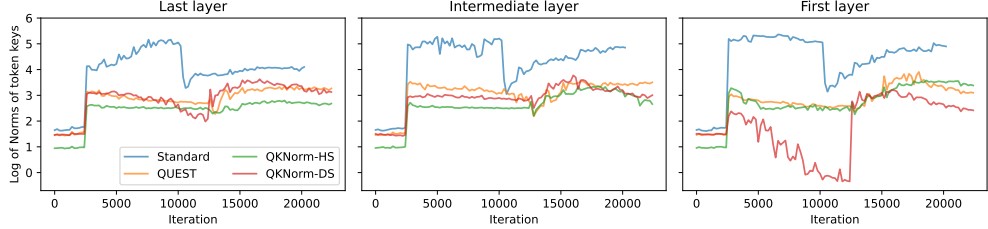

(b) Progression of log of the key norms (corresponding to the maximum logit token above) in the first, intermediate and last layers of a ViT-Base model trained using DeiT and using different attention formulations. For QUEST and QKNorm variants, the norms are prior to the $\ell_2$-normalization and do not have any impact on their attention logits.

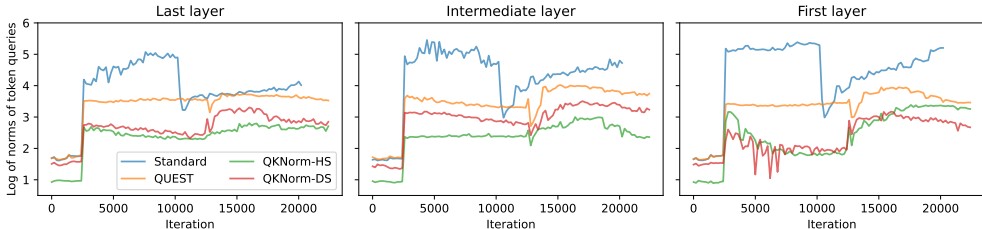

(c) Progression of log of the query norms (corresponding to the maximum logit token above) in the first, intermediate and last layers of a ViT-Base model trained using DeiT and using different attention formulations. For QKNorm variants, the norms are prior to the $\ell_2$-normalization and do not have any impact on their attention logits. Note that the query norms are not $\ell_2$-normalized in QUEST attention.

Figure A6: Maximum logits and their associated query and key norms in different layers for a ViT-Base model trained using DeiT. The model with standard attention crashes around 20000 iterations, which is not observed in the other models. The rapid increase in maximum logits (especially in intermediate and initial layers) is accompanied by an underlying increase in the corresponding query and key norms. Both maximum logits and query norms are stable for QUEST attention, showing that it is not necessary to normalize both queries and keys to ensure training stability.

whereas we were able to train with QUEST attention without any training issues. We find QUEST attention to perform better than standard attention, showing that QUEST attention can also be useful in cross-attention setups. This also demonstrates the potential for QUEST attention to be used as a drop-in replacement for standard attention, orthogonally with other Transformer developments.

Table A1: Ablation of different QK normalization methods for ImageNet classification (ViT-Tiny model trained using DeiT for 300 epochs) [†Liu et al. (2022), ‡Dehghani et al. (2023)]. Means and standard deviations of results are reported over 3 independent runs with different seeds.

| Attention | Scaling | ImageNet-val | | |
|---|---|---|---|---|
| | | Top-1 (%) | Top-5 (%) | NLL $\downarrow$ |
| Standard | $1/\sqrt{D_h}$ | $72.65 \pm 0.10$ | $91.49 \pm 0.03$ | $1.187 \pm 0.003$ |
| QUEST | - | $\mathbf{73.36 \pm 0.16}$ | $\mathbf{91.86 \pm 0.06}$ | $\mathbf{1.160 \pm 0.004}$ |
| QNorm | - | $72.73 \pm 0.11$ | $91.44 \pm 0.02$ | $1.188 \pm 0.004$ |
| QKNorm-HS | $^\dagger \boldsymbol{C} \in \mathbb{R}^{L \times H}$ | $72.49 \pm 0.10$ | $91.43 \pm 0.09$ | $1.198 \pm 0.009$ |
| QKNorm-DS | $^\ddagger \boldsymbol{C}_q, \boldsymbol{C}_k \in \mathbb{R}^{L \times D_h}$ | $71.87 \pm 0.18$ | $90.96 \pm 0.11$ | $1.231 \pm 0.008$ |

Table A2: ImageNet classification validation accuracies with DeiT-Tiny trained using different amounts of training data. Results are averaged over 3 different training data subsets.

| Data (%) | Attention | ImageNet-val | | |
|---|---|---|---|---|
| | | Top-1 | Top-5 | NLL |
| 5% | Standard | $40.82 \pm 0.81$ | $63.92 \pm 0.98$ | $3.326 \pm 0.074$ |
| | QUEST | $\mathbf{41.96 \pm 0.15}$ | $\mathbf{65.24 \pm 0.07}$ | $\mathbf{3.254 \pm 0.014}$ |
| 10% | Standard | $56.02 \pm 0.58$ | $78.52 \pm 0.66$ | $2.201 \pm 0.042$ |
| | QUEST | $\mathbf{57.51 \pm 0.33}$ | $\mathbf{79.75 \pm 0.17}$ | $\mathbf{2.136 \pm 0.016}$ |
| 25% | Standard | $68.87 \pm 0.45$ | $88.86 \pm 0.25$ | $1.386 \pm 0.020$ |
| | QUEST | $\mathbf{69.96 \pm 0.26}$ | $\mathbf{89.50 \pm 0.11}$ | $\mathbf{1.337 \pm 0.006}$ |
| 100% | Standard | $72.50$ | $91.45$ | $1.190$ |
| | QUEST | $\mathbf{73.33}$ | $\mathbf{91.91}$ | $\mathbf{1.160}$ |

Table A3: Learning rate sensitivity of DeiT training evaluated using a ViT-Small model on ImageNet validation performance

| Learning rate | Epochs | Standard attention | | QUEST attention | |
|---|---|---|---|---|---|
| | | Top-1 | Top-5 | Top-1 | Top-5 |
| $1e-4$ | 50 | 62.8 | 84.9 | 63.9 | 85.7 |
| $2e-4$ | 50 | 66.7 | 87.9 | 67.8 | 88.5 |
| $5e-4$ | 50 | $\mathbf{66.9}$ | $\mathbf{87.9}$ | $\mathbf{68.5}$ | $\mathbf{88.7}$ |
| $1e-3$ | 50 | –training crashed– | | 67.9 | 88.5 |
| $2e-3$ | 50 | –training crashed– | | 67.0 | 87.8 |

### A.5.2 ROBUSTNESS IN IMAGE CLASSIFICATION

We evaluate the adversarial robustness of our models trained using DeiT and DeiT-3 using different adversarial attacks using the experimental protocol of Nielsen et al. (2024) and their public codebase[2]. We consider FGSM, PGD and Auto attacks with a perturbation budget of $1/255$ and the SPSA attack with a perturbation budget of 0.1 (under $l_\infty$-norm).

Prior works on adversarial robustness mainly benchmarked on the ViT-Tiny model. We also evaluated larger ViT models to compare standard and QUEST attention. These results are reported in Table A5.

---

[2]https://github.com/stefvk/Elliptical-Attention/tree/main/ImageAttack

Table A4: CrossViT ImageNet Classification

| Model | Attention | Epochs | ImageNet-val | | |
|-------|-----------|--------|--------------|--------|--------|
| | | | Top-1 | Top-5 | NLL |
| CrossViT-9-Dagger | Standard | 300 | 76.4 | 93.4 | 1.009 |
| CrossViT-9-Dagger | QUEST | 300 | 77.0 | 93.6 | 0.991 |
| CrossViT-Small | Standard | 300 | 80.5 | 95.5 | 0.834 |
| CrossViT-Small | QUEST | 300 | 80.9 | 95.6 | 0.826 |
| CrossViT-18-Dagger | Standard | 300 | –training crashed– | | |
| CrossViT-18-Dagger | QUEST | 300 | 82.8 | 96.1 | 0.780 |

For this evaluation, we use the DeiT and DeiT-3 models described in Section 4.1.1 (the clean data results for these models are shown in Table 2). We do not consider Elliptical-QUEST for these larger ViTs as Nielsen et al. (2024) only experimented with ViT-Tiny models and optimal training recipes for larger models are unavailable.

**Elliptical-QUEST:** Elliptical attention proposes to compute the attention logits as $QMK^T/\sqrt{D_H}$, where $M$ is a Mahalanobis factor. The matrix $M$ is a diagonal matrix, where the diagonal elements scale the different feature dimensions in the QK product. In Elliptical-QUEST attention, we obtain the logits as $S\bar{Q}M\bar{K}^T$, where $S$ denotes a diagonal matrix containing the query token norms and $\bar{Q}, \bar{K}$ denote the $\ell_2$-normalized queries and keys respectively. In QUEST attention, the product $\bar{Q}\bar{K}^T$ represents cosine similarity. In Elliptical QUEST, the metric can be interpreted as an elliptical analogue, where each dimension is weighted differently (according to $M$) in a cosine similarity.

Table A5: Robustness to adversarial attacks using larger ViT models trained using DeiT. Note that prior works like Elliptical attention only considered ViT-Tiny models for adversarial robustness evaluation [† DeiT-3 ].

| Model | Attention | Epochs | FGSM | | PGD | | Auto | |
|-------|-----------|--------|-------|------|-------|------|-------|------|
| | | | Top-1 | NLL | Top-1 | NLL | Top-1 | NLL |
| ViT-S/16 | Standard | 200 | 65.90 | 1.362 | 54.47 | 2.050 | 38.22 | 2.699 |
| ViT-S/16 | QUEST | 200 | **67.02** | **1.328** | **57.51** | **1.886** | **40.98** | **2.540** |
| ViT-B/16[†] | Standard | 400 + 20 | 69.41 | 1.229 | 52.55 | 2.080 | – | – |
| ViT-B/16[†] | QUEST | 400 + 20 | **70.67** | **1.194** | **54.64** | **2.033** | – | – |
| ViT-L/16 | QKNorm-LN | 100 | 58.95 | 1.849 | 51.79 | 2.355 | – | – |
| ViT-L/16 | QUEST | 100 | **62.85** | **1.669** | **53.54** | **2.316** | – | – |

### A.5.3 IMAGE CLASSIFICATION - EXPLAINABILITY

We evaluate explainability of a model using a recent method, AG-CAM (Leem & Seo, 2024) that combines attention maps with gradient information. We adapt the code from their public repository[3] to DeiT-3 models. Based on the ViT-B model trained using DeiT-3, we show additional qualitative examples in Figures A7 and A8.

### A.5.4 IMAGE SEGMENTATION

The image segmentation models are trained on the ADE20K dataset (Zhou et al., 2019; 2017) following the setup of Segmenter (Strudel et al., 2021). We use the same training configurations as in Nielsen et al. (2024) for both ViT-Ti and ViT-S backbones. The ViT backbones are initialized with the weights obtained from the DeiT trainings in Section 4.1.1 and then, we finetune the entire model, both the encoder and the decoder. This experiment is conducted on a single NVIDIA A100 40GB GPU. We train these segmentation models for 160K iterations with a global batch size of 8. An SGD optimizer with a starting learning rate of 0.001 and polynomial learning rate scheduling is used.

---

[3]https://github.com/LeemSaebom/Attention-Guided-CAM-Visual-Explanations-of-Vision-Transformer-Guided-by-Self-Attention

Table A6: DeiT-3 ImageNet Classification with large-scale ViT models

| Model | Parameters | Attention | Epochs | Top-1 | IN-val Top-5 | NLL ↓ |
|-------|-----------|-----------|--------|-------|--------------|-------|
| ViT-H/14 | 600M | Standard | 100 | 68.3 | 88.8 | 1.545 |
| ViT-H/14 | 600M | QUEST | 100 | **69.2** | **89.4** | **1.495** |
| ViT-H/14 | 600M | Standard | 100 + 20 | 76.2 | 93.3 | 1.042 |
| ViT-H/14 | 600M | QUEST | 100 + 20 | **76.8** | **93.6** | **1.002** |
| ViT-2B/16 | 2.4B | Standard | 10 | 3.1 | 9.4 | 6.153 |
| ViT-2B/16 | 2.4B | QKNorm-DS | 10 | 5.2 | 14.9 | 5.740 |
| ViT-2B/16 | 2.4B | QUEST | 10 | **11.9** | **27.8** | **5.011** |

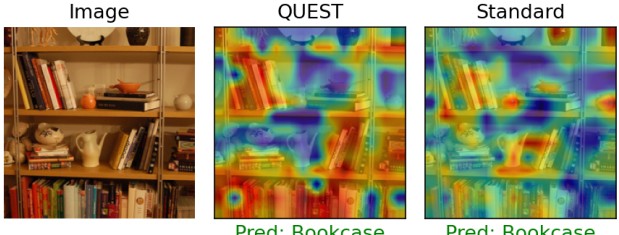

(a) Example from the "Bookcase" class. QUEST attention focuses similarly on most of the books and shelves. Standard attention focuses only a few of the instances.

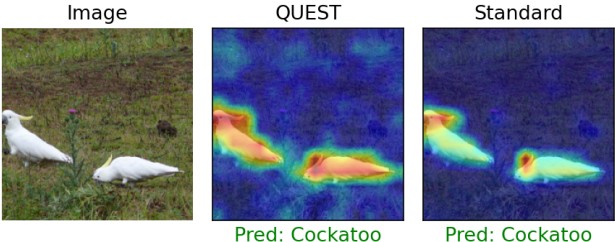

(b) Example from the "Cockatoo" class. Standard attention only focuses on a specific part of the birds. QUEST attention evenly attends to the entire birds.

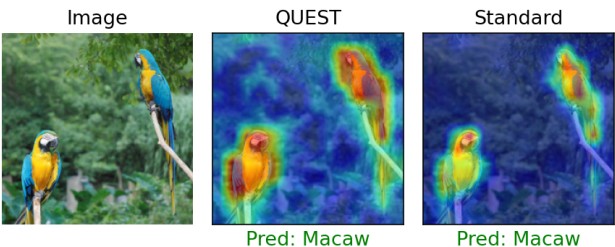

(c) Example from the "Macaw" class. Standard attention only focuses on a specific part of the birds. QUEST attention evenly attends to the entire birds.

Figure A7: Examples showing class activation maps (CAM) for images from the ImageNet dataset. The CAM is obtained with the AG-CAM method using the DeiT-3 models shown in 4.1.1. Standard attention concentrates attention on specific parts of the object. On the other hand, QUEST attention attends more evenly to the entire object or all relevant regions.

### A.5.5 LANGUAGE MODELING

We follow the experimental protocol of Nguyen et al. (2022b); Nielsen et al. (2024); Schlag et al. (2021) and train Transformer models of Small (44M parameters) and Medium (90M parameters) sizes on the clean WikiText-103 dataset (Merity et al., 2017). The models are trained using Adam and using the code and hyperparameters from the Elliptical Attention repository [4]. The Transformer-Small model is trained for 120 epochs with a batch size of 96, starting learning rate of 0.00025 and cosine scheduling. The Transformer-Medium model is trained with a batch size of 56, starting learning rate of 0.00025 and cosine scheduling. Following the standard evaluation setting in Schlag et al. (2021), we process the text sequence using a sliding window (256 for Transformer-Small and 384 for Transformer-Medium). The perplexity is computed based on the last position except for the first segment, where it is evaluated for all positions. We also experiment with the larger Transformer-XL-Standard (151M parameters) model (note that the model is called Base in the repository) from Dai et al. (2019) using their public codebase [5]. We use the hyperparameter setup from the repository and this uses the same evaluation protocol described above.

### A.5.6 TIME SERIES CLASSIFICATION

We use the training and evaluation protocol from the Time Series Library (TSLib) repository [6]. The default Transformer model in the repository concatenates the output token features and uses a linear classification layer. The final classification layer can become arbitrarily large since the weights $W_{\text{CLS}} \in \mathbb{R}^{ND \times C}$ where $N$ is the sequence length, $D$ is the embedding dimensionality and $C$ is the number of classes. Instead, we found that using a [CLS] token consistently improved the standard Transformer results (see Table A7). Hence, we use Transformers with a [CLS] token and the classification layer only depends on the CLS token output from the Transformer. Both standard and QUEST attention use this same setup in our experiments. We use the same training hyperparameters for both attention types, based on the default configuration for a standard Transformer provided in the repository. This consists of training a 3-layer Transformer with a model dimension of 128, a batch size of 16, a learning rate of 0.001 using the RAdam (Liu et al., 2020a) optimizer for 100 epochs (with an early stopping patience of 10 epochs). In Table A8, we show additional ablation experiment comparing with QNorm and QKNorm attention.

Table A7: Impact of using a [CLS] token for UEA Multivariate Time Series Classification using a standard Transformer

| Dataset | Standard | Standard + [CLS] |
|---|---|---|
| EthanolConcentration | 28.14 | **29.28** |
| FaceDetection | **67.99** | 65.24 |
| Handwriting | 38.24 | **42.00** |
| Heartbeat | 77.07 | **77.56** |
| JapaneseVowels | 97.84 | **98.38** |
| PEMS-SF | **84.97** | 83.82 |
| SelfRegulationSCP1 | **90.44** | 88.05 |
| SelfRegulationSCP2 | 55.00 | **58.89** |
| SpokenArabicDigits | 98.41 | **98.86** |
| UWaveGestureLibrary | 86.25 | **86.88** |
| Average | 72.44 | **72.90** |

### A.5.7 POINTCLOUD SEGMENTATION

We consider the pointcloud segmentation using PointTransformer-V3 model architecture (Wu et al., 2024) and use the Pointcept (Contributors, 2023) training framework. For this task, we use the nuScenes (Caesar et al., 2020) dataset with the same training configurations as in the Pointcept

---

[4]https://github.com/stefvk/Elliptical-Attention/tree/main/Wikitext

[5]https://github.com/kimiyoung/transformer-xl

[6]https://github.com/thuml/Time-Series-Library

Table A8: Ablation experiment for UEA Multivariate Time Series Classification using different forms of attention

| Dataset | Standard | QNorm | QKNorm | QUEST |
|---|---|---|---|---|
| EthanolConcentration | 29.28 | 29.28 | 29.28 | **30.42** |
| FaceDetection | 65.24 | 64.81 | 64.73 | **65.83** |
| Handwriting | 42.00 | 41.06 | **49.18** | **49.18** |
| Heartbeat | 77.56 | 75.61 | 77.56 | **78.54** |
| JapaneseVowels | 98.38 | **98.92** | 98.38 | 98.38 |
| PEMS-SF | **83.82** | 80.92 | 79.19 | 80.92 |
| SelfRegulationSCP1 | 88.05 | 87.71 | **88.76** | 88.05 |
| SelfRegulationSCP2 | **58.89** | 56.67 | 54.44 | 58.33 |
| SpokenArabicDigits | 98.86 | 99.41 | **99.45** | 99.41 |
| UWaveGestureLibrary | **86.88** | 85.00 | 86.81 | 86.25 |
| Average | 72.90 | 71.94 | 72.78 | **73.53** |

repository. We train PointTransformer-V3 models using standard, QKNorm and QUEST attentions for 50 epochs and the validation mIoU scores are reported in Table A9.

Table A9: Pointcloud segmentation using PointTransformer-V3 model on the nuScenes dataset. We report the mIoU on the validation set.

| Model | Parameters | Attention | nuScenes val. mIoU (%) |
|---|---|---|---|
| PointTransformer-V3 | 46.2 M | Standard | 80.40 |
| PointTransformer-V3 | 46.2 M | QKNorm-DS | 80.37 |
| PointTransformer-V3 | 46.2 M | QUEST | **80.83** |

### A.5.8 GRAPH TRANSFORMER BENCHMARKS

We consider the ZINC, CIFAR10, PATTERN and CLUSTER tasks from the standard Graph Neural Network (GNN) benchmarks (Dwivedi et al., 2023), detailed as follows:

- **ZINC** is a regression task for a molecular property and it is evaluated using the Mean Absolute Error (MAE).
- **CIFAR10** is a graph classification task based on superpixel graphs and it is evaluated using classification accuracy.
- **CLUSTER** and **PATTERN** are node classification tasks and they are evaluated using class-weighted classification accuracy.

From the long-range graph benchmarks (Dwivedi et al., 2022), we consider the PascalVOC-SP, COCO-SP, Peptides-func, Peptides-struct and PCQM-Contact tasks, detailed as follows:

- **PascalVOC-SP** and **COCO-SP** are node classification tasks based on the Pascal-VOC dataset (Everingham et al., 2010) and the MS-COCO (Lin et al., 2014) datasets respectively. For both tasks, the macro weighted F1 score is used as the performance metric.
- **PCQM-Contact** is a link prediction task and it is evaluated using the Mean Reciprocal Rank (MRR) (Hoyt et al., 2022).
- **Peptides-func** is a multi-label classification task with 10 classes and it is evaluated using the unweighted mean Average Precision (AP).
- **Peptides-struct** is a multi-label regression task for graph-level properties and it is evaluated using the Mean Absolute Error (MAE).

The hyperparameter setups to train GPS models for standard GNN benchmarks is shown in Table A10. Similarly, the hyperparameter setups to train GPS models for long-range graph benchmarks is shown

in Table A11. The Graph Transformers were trained and evaluated using the GraphGPS repository [7] and the shared configuration files for each dataset.

Table A10: GPS hyperparameter setup for standard GNN benchmarks (Dwivedi et al., 2023)

| Hyperparameter | ZINC | CIFAR10 | PATTERN | CLUSTER |
|---|---|---|---|---|
| # GPS Layers | 10 | 3 | 6 | 16 |
| Hidden dim | 64 | 52 | 64 | 48 |
| GPS-MPNN | GINE | GatedGCN | GatedGCN | GatedGCN |
| GPS-GlobAttn | Transformer | Transformer | Transformer | Transformer |
| # Heads | 4 | 4 | 4 | 8 |
| Dropout | 0 | 0 | 0 | 0.1 |
| Attention dropout | 0.5 | 0.5 | 0.5 | 0.5 |
| Graph pooling | sum | mean | – | – |
| Positional Encoding | RWSE-20 | LapPE-8 | LapPE-16 | LapPE-10 |
| PE dim | 28 | 8 | 16 | 16 |
| PE encoder | linear | DeepSet | DeepSet | DeepSet |
| Batch size | 32 | 16 | 32 | 16 |
| Learning Rate | 0.001 | 0.001 | 0.0005 | 0.0005 |
| # Epochs | 2000 | 100 | 100 | 100 |
| # Warmup epochs | 50 | 5 | 5 | 5 |
| Weight decay | 1e-5 | 1e-5 | 1e-5 | 1e-5 |
| # Parameters | 423,717 | 112,726 | 337,201 | 502,054 |
| PE precompute | 23s | 2.55min | 28s | 67s |
| Time (epoch/total) | 21s / 11.67h | 64s / 1.78h | 32s / 0.89h | 86s / 2.40h |

Table A11: GPS hyperparameter setup for long-range graph benchmarks (Dwivedi et al., 2022)

| Hyperparameter | PascalVOC-SP | COCO-SP | PCQM-Contact | Peptides-func | Peptides-struct |
|---|---|---|---|---|---|
| # GPS Layers | 4 | 4 | 4 | 4 | 4 |
| Hidden dim | 96 | 96 | 96 | 96 | 96 |
| GPS-MPNN | GatedGCN | GatedGCN | GatedGCN | GatedGCN | GatedGCN |
| GPS-SelfAttn | Transformer | Transformer | Transformer | Transformer | Transformer |
| # Heads | 8 | 8 | 4 | 4 | 4 |
| Dropout | 0 | 0 | 0 | 0 | 0 |
| Attention dropout | 0.5 | 0.5 | 0.5 | 0.5 | 0.5 |
| Graph pooling | – | – | – | mean | mean |
| Positional Encoding | LapPE-10 | LapPE-10 | LapPE-10 | LapPE-10 | LapPE-10 |
| PE dim | 16 | 16 | 16 | 16 | 16 |
| PE encoder | DeepSet | DeepSet | DeepSet | DeepSet | DeepSet |
| Batch size | 32 | 32 | 256 | 128 | 128 |
| Learning Rate | 0.0005 | 0.0005 | 0.0003 | 0.0003 | 0.0003 |
| # Epochs | 300 | 300 | 200 | 200 | 200 |
| # Warmup epochs | 10 | 10 | 10 | 5 | 5 |
| Weight decay | 0 | 0 | 0 | 0 | 0 |
| # Parameters | 510,453 | 516,273 | 512,704 | 504,362 | 504,459 |
| PE precompute | 8.7min | 1h 34min | 5.23mi | n 73s | 73s |
| Time (epoch/total) | 17.5s / 1.46h | 213s / 17.8h | 154s / 8.54h | 6.36s / 0.35h | 6.15s / 0.34h |

### A.5.9 SELF-SUPERVISED LEARNING

We also conduct a self-supervised learning (SSL) experiment using the iBOT (Zhou et al., 2021) method (which is the foundation for SOTA SSL models like DINOv2 (Oquab et al., 2024)). This experiment was ran on a single node of 8 NVIDIA A100 40GB GPUs. When we attempted to reproduce DINOv2 pre-training on ImageNet-1K using the ViT-Large model (standard attention)

---

[7]https://github.com/rampasek/GraphGPS/

with a smaller batch size of 512 (instead of 2048), we observed a significant drop in performance to 68.2% kNN accuracy (vs 81.6 % reported in their repository). Since training DINOv2 with smaller batch sizes was non-trivial, we instead opted for the iBOT method. We detail the pre-training and downstream evaluations below.

**Pre-training:** We pre-train a ViT-Base/16 model using the iBOT (Zhou et al., 2021) method and the vMF normalization (Govindarajan et al., 2023) for 400 epochs on the ImageNet-1K dataset (Deng et al., 2009) using the code from the iBOT repository [8] and with the exact same hyperparameters as in iBOT. The pre-training is carried out on a single node consisting of 8 NVIDIA A100 40GB GPUs.

**ImageNet downstream tasks:** We evaluate the kNN performance using the same protocol as DINO (Caron et al., 2021). We use weighted k-NN (temperature = 0.07) and report the best result among those obtained with $k = \{5, 10, 20, 100, 200\}$. We generally found $k = 10$ to produce the best result for both standard and QUEST attention. For linear classification, we follow the evaluation protocol of iBOT (Zhou et al., 2021) and report the best results among those obtained using different learning rates. The linear classifier is trained on the features obtained by concatenating the [CLS] features and average pooling of the patch features. The training is run for 100 epochs using the same hyperparameter setup as in iBOT. For evaluations with limited training data, we follow the evaluation protocol of Assran et al. (2022) and use the same data subsets. We report the average validation accuracy over 3 different data subsets for the 1, 2 and 5 images per class settings. For finetuning on ImageNet, we train for 100 epochs with a layer-wise learning rate decay of 0.65, following the effective recipe from BeIT (Bao et al., 2022). We report the best performance after considering different learning rates from $\{8e-4, 9e-4, 1e-3, 2e-3\}$. These results on ImageNet downstream tasks are reported in Table A12. We observe consistent improvements on all the considered settings. Finetuning models from SSL pre-trained weights is common in recent times and we highlight that QUEST attention can also bring improvements in this setting (84.4 % vs 84.1 %).

**Transfer linear probing:** We conduct transfer linear probing evaluation by freezing the pre-trained model and training a linear classifier on the [CLS] features output by the model. We follow the evaluation protocol of Ericsson et al. (2021) and Chen et al. (2020) and train $\ell_2$-regularized linear classifiers. We select the regularization strength among a set of 45 values spaced linearly in the range $[-6, 5]$ in log-space and compute the standard evaluation metric for each dataset. The dataset suite includes the following datasets: Aircraft (Maji et al., 2013), Caltech101 (Li et al., 2022b), Describable Textures Dataset (DTD) (Cimpoi et al., 2014), Flowers (Nilsback & Zisserman, 2008), Food (Bossard et al., 2014), Pets (Parkhi et al., 2012) and SUN397 (Xiao et al., 2010; 2016) datasets. The detailed transfer linear probing results are shown in Table A14. QUEST performs significantly better than standard attention on Aircraft and Flowers datasets. On other datasets, the results are somewhat mixed. Nevertheless, QUEST performs better than standard attention in terms of the overall average.

Table A12: Self-supervised pre-training on ImageNet with iBOT and evaluating on ImageNet tasks

| Attention | kNN | Linear | Finetuning | Few-shot | | | |
|---|---|---|---|---|---|---|---|
| | | | | 1 img/cls | 2 imgs/cls | 5 imgs/cls | 1% imgs |
| Standard | 78.7 | 80.3 | 84.1 | $51.6 \pm 0.1$ | $61.1 \pm 0.7$ | $68.3 \pm 0.3$ | 72.3 |
| QUEST | **79.0** | **80.5** | **84.4** | $\mathbf{52.5 \pm 0.2}$ | $\mathbf{61.7 \pm 0.6}$ | $\mathbf{69.0 \pm 0.1}$ | **72.7** |

Table A13: Self-supervised pre-training on ImageNet with iBOT and evaluating on transfer tasks

| Attention | Transfer linear probe | Image Retrieval | | | | VOS | | |
|---|---|---|---|---|---|---|---|---|
| | Average | RParis | | ROxford | | DAVIS-2017 | | |
| | | M | H | M | H | $\mathcal{J}\&\mathcal{F}_m$ | $\mathcal{J}_m$ | $\mathcal{F}_m$ |
| Standard | 81.5 | 65.4 | 38.1 | 38.1 | 13.8 | 63.1 | **61.9** | 64.2 |
| QUEST | **81.9** | **65.8** | **39.3** | **38.6** | **15.3** | **63.2** | 61.8 | **64.5** |

---

[8] https://github.com/bytedance/ibot/

Table A14: Self-supervised pre-training on ImageNet-1K with iBOT-vMF and evaluating with transfer linear probes

| Attention | Acft. | Cal101 | DTD | Flwrs. | Food | Pets | SUN | Avg. |
|---|---|---|---|---|---|---|---|---|
| Standard | 58.1 | **95.5** | **74.7** | 94.8 | 83.6 | 93.9 | **70.2** | 81.5 |
| QUEST | **59.9** | 95.1 | 74.4 | **95.8** | **83.9** | **94.3** | 69.8 | **81.9** |

**Other transfer learning tasks:** For image retrieval experiments, we follow the evaluation protocol of DINO and evaluate on the face-blurred versions (v1.0) of the Oxford and Paris datasets. We perform image retrieval based on nearest neighbors and report the mean Average Precision (mAP) on the medium (M) and hard (H) data splits for each dataset. For video object segmentation (VOS), we use the evaluation protocol of DINO and evaluate VOS performance using standard metrics on the DAVIS-2017 benchmark dataset (Pont-Tuset et al., 2017). These transfer learning results are reported in Table A13. We find that QUEST attention performs better on image retrieval and similar to standard attention on video object segmentation.

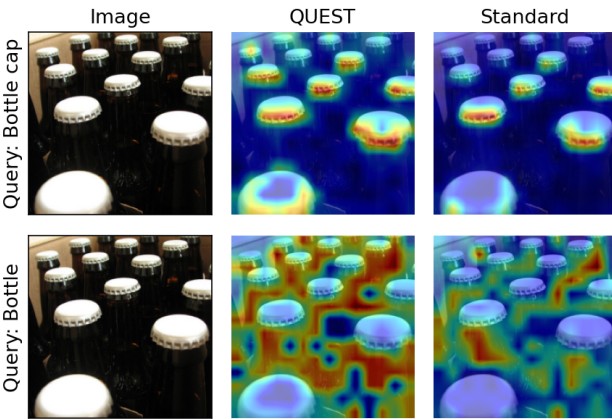

(a) Image queried with the target class of Bottle and Bottlecap.

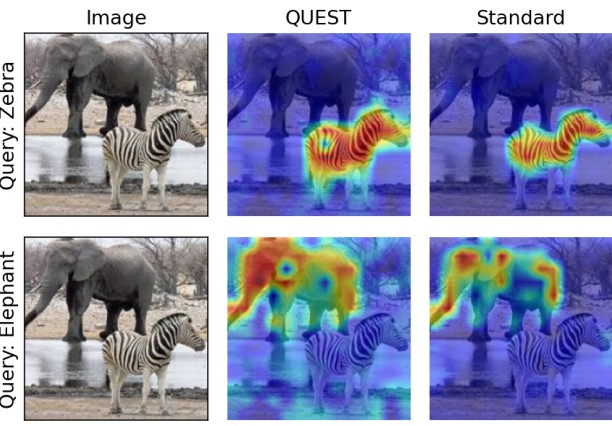

(b) Image queried with the target class of Elephant and Zebra.

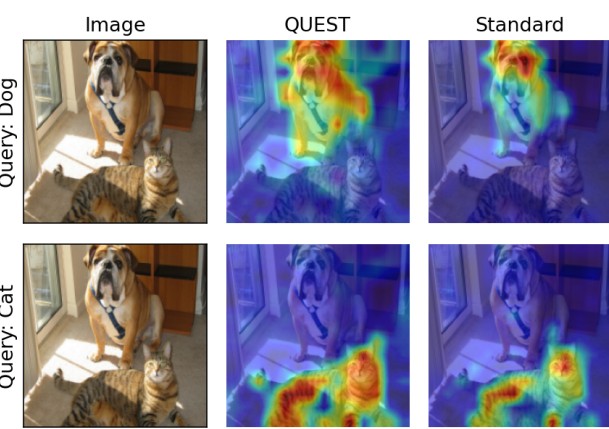

(c) Image queried with the target class of Dog and Cat.

Figure A8: Examples showing class activation maps (CAM) for images containing two distinct objects. The CAM is obtained with the AG-CAM method by querying for the specified classes (i.e. using the specified class as the target) using the DeiT-3 models shown in 4.1.1. Standard attention concentrates attention on specific parts of the object or fewer instances of the object. On the other hand, QUEST attention attends more evenly to the entire object or all relevant regions.

