# OpenReview forum: "QUEST: A robust attention formulation using query-modulated spherical attention"
_ICLR.cc/2026/Conference — ICLR 2026 Poster_

### Official Review · Reviewer_5T4g · 2025-10-23

**Soundness:** 2
**Presentation:** 2
**Contribution:** 2
**Rating:** 2
**Confidence:** 4

**Summary:**

This paper analyzes the role of the norm of queries and keys in the attention module, proposes an alternative variant to the standard attention mechanism, and demonstrates the effectiveness of the proposed attention variant across multiple domains.

**Strengths:**

1. The designed attention variant serves as a drop-in replacement for the standard approach, requiring only simple modifications.

2. Experiments were conducted across multiple domains, including image classification, language modeling, Graph Transformers, and time series tasks.

**Weaknesses:**

see questions.

**Questions:**

1.**Question regarding Fig. 1**. In the third row of Fig. 1, the standard ViT attends to the bird, yet it still makes an incorrect prediction. I hope the authors can provide a clear explanation to help readers understand this phenomenon.

2.**The baselines in Table 2 appear relatively low**. First, when using DeiT and MAE recipes for supervised training standard ViT-Base/Large on ImageNet, I did not encounter training instability issues. Second, using the DeiT-3 training method (fourth row), the authors report a baseline accuracy of 82.7%, which is lower than the official DeiT-3 result of 83.8% (which I have successfully reproduced). Third, I request the authors to provide the training curves.

3.**Lack of large-scale experiments**. Although the effectiveness of the proposed method has been validated across multiple domains (cf. Tables 2, 5, 10), the model parameters used are relatively small. As a simple replacement for standard attention, I believe more large-scale experiments should be conducted.

---

> ### Author Response · Authors · 2025-11-23
> **Response to Reviewer 5T4g (part 1/3)**
>
> We thank the reviewer for their valuable feedback. We address the weaknesses and questions raised by the reviewer below, in addition to the general rebuttal response.
>
> > Q1. Question regarding Fig. 1. In the third row of Fig. 1, the standard ViT attends to the bird, yet it still makes an incorrect prediction. I hope the authors can provide a clear explanation to help readers understand this phenomenon.
>
> The class activation maps are a visual tool to understand which part of the input the model attends to, but it's important to keep in mind that this does not give the complete picture of what's going on inside the transformer when interpreting the results. Specifically, attending to the correct image region should be seen as an indication that the model has identified the correct object, but it is not a guarantee that the classification will be correct (not that in our example, the model misclassifies the image as another bird species, so it's natural that it still attends to the bird).
>
> In row 1, standard attention attends to the birds in the top half of the image but not the other birds. This shows that the birds in the bottom half of the image do not contribute to this correct prediction. In row 3, when the top part of the image is noised, the model focuses on the birds in the bottom part of the image (since they are the most salient object in the image now). Then, it becomes clear that the model associates the birds in the bottom part of the image to a spoonbill. This demonstrates that standard attention learned to only focus on some of the bird instances to arrive at the correct prediction but misclassifies when forced to predict using the other bird instances. We will update the image caption to explain this more clearly when we revise the paper.

---

> ### Author Response · Authors · 2025-11-23
> **Response to Reviewer 5T4g (part 2/3)**
>
> > Q2. The baselines in Table 2 appear relatively low. First, when using DeiT and MAE recipes for supervised training standard ViT-Base/Large on ImageNet, I did not encounter training instability issues. Second, using the DeiT-3 training method (fourth row), the authors report a baseline accuracy of 82.7%, which is lower than the official DeiT-3 result of 83.8% (which I have successfully reproduced). Third, I request the authors to provide the training curves.
>
> [1, 2] showed that attention entropy collapse or attention logit explosion can indeed occur in baseline Transformer benchmarks such as DeIT at model sizes as small as ViT-Base. This is generally observed when even slightly higher learning rates are used. The observed training crashes pertain to the DeIT training recipe and are also observed by several others who have attempted to reproduce the ViT-Base experiment (see Github issue https://github.com/facebookresearch/deit/issues/29 which is also referenced by many others facing similar issues). Note that DeIT3 and MAE use improved training recipes compared to DeIT. MAE uses layer-wise learning rate decay which is useful when training deeper models. DeIT3 uses layer scale, FixRes regularization and better data augmentation. The fine-tuning in the self-supervised learning experiment (see section A.4.8) uses a fine-tuning recipe based on BeIT that includes layer-wise learning rate decay. We find QUEST to perform better than the standard attention baseline, showing that QUEST can benefit from these orthogonal improvements.
>
> Firstly, we would like to clarify that the DeIT3 training reported in Table 2, row 4 is based on a 400+20 epoch training. The corresponding performance reported in DeIT-3 is 83.5%. We use the public repository to run this experiment and using the hyperparameter setup in the command provided in the repo for 800 epoch training. However, we use a recent Pytorch v2.1.0 instead of the older Pytorch 1.13.1 (in their repo) for all DeIT experiments. We used the LAMB optimizer instead of FusedLAMB, which is only an efficient implementation. Another possible difference could be that some hyperparameter was different for the 400 epoch training. In appendix B of DeIT3, it is mentioned that a weight decay of 0.02 was used for shorter trainings. Hence, it is not fully clear if the 400 epoch training used the exact training setup as the 800 epoch training. We provide the exact command used for this training below.
>
> We conducted two experiments to further investigate this. First, we repeated the DeIT3 ViT-Base 400+20 epoch experiment without automatic mixed precision (AMP). The default on the DeIT repo is to train with AMP but some users had noted better performance without AMP on DeIT. We found roughly similar results as reported previously. Standard attention performs marginally better (82.9 %) and QUEST attention achieved the same performance (83.2 %). So, our observation that QUEST performed better still holds true. Second, we also conducted shorter 100+20 epoch experiments using a smaller weight decay of 0.02. We still found QUEST attention to outperform standard attention (see table below).
>
> | Attention | Epochs | Val. Accuracy (%) |
> |---|---|---|
> | Standard | 100 | 72.9 |
> | QUEST | 100 | 74.2 |
> | Standard | 100+20 | 75.0 |
> | QUEST | 100+20 | 76.1 |
>
> All the reported results in the paper are based on our own experiments using the same training setup and environment. We expect this to ensure that the comparisons we make are still fair, even if some other underlying aspect might have changed.
>
> We have added the training curves for DeIT-3 ViT-Base in the supplementary material. Note that the different loss values between train, test and fine-tuning comes from the different losses being used in DeIT. But, the test loss in both phases is the standard CrossEntropy loss. We also added the exact commands specifying the hyperparameter setup in the supplementary material.
>
> [1] Zhai et al. Stabilizing transformer training by preventing attention entropy collapse. In ICML, 2023.
>
> [2] Wortsman et al. Small-scale proxies for large-scale transformer training instabilities. In ICLR, 2024.

---

> ### Author Response · Authors · 2025-11-23
> **Response to Reviewer 5T4g (part 3/3)**
>
> > Q3. Lack of large-scale experiments. Although the effectiveness of the proposed method has been validated across multiple domains (cf. Tables 2, 5, 10), the model parameters used are relatively small. As a simple replacement for standard attention, I believe more large-scale experiments should be conducted.
>
> While the modification is simple, training larger models also require a much larger training compute which we do not have currently. We have run additional experiments using ViT-Huge (600M parameters) and a short 10 epoch training with ViT-2B (2.4B parameters) which also demonstrate favorable results for QUEST. We train these models using the DeIT-3 training recipe. QUEST is stable to train without lowering the learning rates even at the 2B parameter scale (we use a learning rate of 1e-3).
>
> | Model | Parameters | Attention | Epochs | Top-1 | Top-5 | NLL $\downarrow$ |
> |---|---|---|---|---|---|---|
> |ViT-H/16 | 600M | Standard | 100 | 68.3 | 88.8 | 1.545 |
> |ViT-H/16 | 600M | QUEST | 100 | 69.2 | 89.4 | 1.495 |
> |ViT-H/16 | 600M | Standard | 100 + 20 | 76.2 | 93.3 | 1.042 |
> |ViT-H/16 | 600M | QUEST | 100 + 20 | 76.8 | 93.6 | 1.002 |
> |ViT-2B/16 | 2.4B | Standard | 10 | 3.1 | 9.4 | 6.153 |
> |ViT-2B/16 | 2.4B | QKNorm-DS | 10 | 5.2 | 14.9 | 5.740 |
> |ViT-2B/16 | 2.4B | QUEST | 10 | 11.9 | 27.8 | 5.011 |

---

> ### Comment · Reviewer_5T4g · 2025-11-24
>
> Q1. The author explains that the key to standard attention correctly identifying Macaw lies in its need to focus on the bird instances in the upper half of the image rather than the lower half. I find this explanation unconvincing and lacking direct evidence. **Figure 1 is confusing, as it does not clearly demonstrate the advantages of the proposed QUEST method over standard attention.**
>
> Q2. Taking the ViT-Large model trained from scratch as an example, MAE/DeiT-3 reported accuracies of 82.6% and 84.5%. **The authors should demonstrate that using the proposed QUEST method can achieve results that are comparable or even stronger.**
> However, the authors only showed that QUEST outperforms standard attention over short training epochs (e.g., 100 epochs). It should be noted that with such short training epochs, the model has not yet fully converged! Superior performance over short training epochs does not guarantee the same advantage over longer training epochs!
>
> Q3. I appreciate that the authors have supplemented the results for ViT-Huge and ViT-2B. However, similar to Q2, the authors should demonstrate that using QUEST can achieve results comparable to standard attention, such as 85.1% with ViT-H.

---

> > ### Author Response · Authors · 2025-12-02
> > **Response to Reviewer 5T4g (part 1/2)**
> >
> > We thank the reviewer for their thoughtful feedback on our rebuttal response. We have taken some time to conduct additional longer experiments requested by the reviewer (to the extent possible within our computational resources) and these are detailed in our response below.
> >
> > > Q1. The author explains that the key to standard attention correctly identifying Macaw lies in its need to focus on the bird instances in the upper half of the image rather than the lower half. I find this explanation unconvincing and lacking direct evidence. Figure 1 is confusing, as it does not clearly demonstrate the advantages of the proposed QUEST method over standard attention.
> >
> > Thank you for bringing this up. We do not consider the figure as “evidence” for the improved performance of QUEST, but rather as a visual illustration of how the method differs (in terms of its attention mask) compared to standard attention. We believe that this sheds some light on the inner workings of the attention mechanism and how it is affected by the proposed modification, specifically that it leads to a more diverse attention mask. This can be understood by noting that QUEST normalization prevents specific tokens from “stealing global attention”, as we also discuss in the toy problem.
> >
> > We also find it reasonable that a more diverse attention can make the models more robust to input data variations (which supports our observation of improved adversarial robustness using QUEST in section 4.1.2) and believe that the shown example is an indication of this. But, again, we do not claim that this is a formal proof in any way. The attention mask is just a visual tool for interpreting the model and we do not claim that the decision rule implemented by the trained classifier is as simple as “attending to the bird gives the correct classification”.
> >
> > We personally believe that the figure is useful as a visual illustration, but we are happy to remove it if there is consensus that it is more confusing than helpful. We would appreciate the input from the AC regarding this matter.

---

> > > ### Author Response · Authors · 2025-12-02
> > > **Response to Reviewer 5T4g (part 2/2)**
> > >
> > > > Q2. Taking the ViT-Large model trained from scratch as an example, MAE/DeiT-3 reported accuracies of 82.6% and 84.5%. The authors should demonstrate that using the proposed QUEST method can achieve results that are comparable or even stronger. However, the authors only showed that QUEST outperforms standard attention over short training epochs (e.g., 100 epochs). It should be noted that with such short training epochs, the model has not yet fully converged! Superior performance over short training epochs does not guarantee the same advantage over longer training epochs!
> > >
> > > > Q3. I appreciate that the authors have supplemented the results for ViT-Huge and ViT-2B. However, similar to Q2, the authors should demonstrate that using QUEST can achieve results comparable to standard attention, such as 85.1% with ViT-H.
> > >
> > > We would like to highlight that the experiments using ViT-Tiny and ViT-Base model sizes (see Table 1 and 2 in the paper) are trained for a larger number of epochs (300 for ViT-Tiny and 400+20 for ViT-Base) to ensure that the models converged. We would like to clarify the purpose of the shorter training experiment on ViT-Large, included in Table 1 of the paper. This experiment on the ViT-Large (a shorter training for 100 epochs, see Table 1) model using the **DeIT-1** training recipe was mainly conducted to demonstrate that this training was stable with QUEST (DeIT-1 [1] does not contain a result on ViT-L) while the training crashed with standard attention. Further, we found that QUEST achieved better performance than the stable attention baseline, QKNorm. The performance was measured within a given training budget, where both QUEST as well as the baseline model were trained for 100 epochs.
> > >
> > > We agree with the reviewer that the training might not have fully converged in the shorter trainings that we used earlier for ViT-L (in the paper) and ViT-H (in the earlier rebuttal response). Hence, we have carried out longer trainings for 400+20 epochs for both ViT-L/16 and ViT-H/14 models using the **DeIT-3** training recipe. The DeIT-3 training recipe consists of two phases - training with smaller input image resolution for 400 epochs and then finetuning with standard 224x224 image size for 20 epochs. Given the computational limitations, we made the following modifications to enable these longer trainings:
> > > - **ViT-L**: We used a smaller global batch size of 1024 instead of 2048 as in the original training setting.
> > > - **ViT-H**: We used a smaller global batch size of 1024 instead of 2048 as in the original training setting. We used a smaller input image size of 126x126 in the first training phase, instead of 182x182.
> > >
> > > For larger models, we found batch size reduction to have a negative impact on performance. This results in a reduced performance for ViT-L/16 compared to the results reported in DeIT-3 [2]. ViT-H/14 performance is reduced to a larger extent since we also trained with a smaller input resolution in the first training phase. Nevertheless, these experiments enable us to compare QUEST and standard attention in larger models that are trained until convergence. We observe that QUEST attention still results in slightly improved performance (+0.2%) compared to standard attention. Taking the results in Table 1 and our new experiments during the rebuttal into account, we find QUEST attention to improve performance *consistently* for all model sizes both in longer sufficiently coverged trainings as well as in shorter trainings with limited compute budget.
> > >
> > > | Model | Attention | Epochs | Top-1 Acc (%) |
> > > |---|---|---|---|
> > > | ViT-L/16 | Standard | 400+20 | 83.9 |
> > > | ViT-L/16 | QUEST | 400+20 | **84.1** |
> > > | ViT-H/14 | Standard | 400+20 | 83.2 |
> > > | ViT-H/14 | QUEST | 400+20 | **83.4** |
> > >
> > > [1] Touvron et al. "Training data-efficient image transformers & distillation through attention." ICML, 2021.
> > >
> > > [2] Touvron et al. "DeIT III: Revenge of the ViT." ECCV, 2022.

---

### Official Review · Reviewer_bMvw · 2025-10-25

**Soundness:** 3
**Presentation:** 3
**Contribution:** 2
**Rating:** 6
**Confidence:** 3

**Summary:**

This paper presents QUEST, a technique designed to enhance training stability and feature learning in Attention mechanisms. The authors investigate how attention logit magnitudes and the norms of Query (Q) and Key (K) matrices influence training dynamics. They find that applying global normalization to the K matrix can cause the model to prematurely focus on less significant features during early training phases, leading to suboptimal convergence. Additionally, since K's norm influences gradient magnitudes, this approach creates training instability. These findings are supported by prior research and demonstrated through a simplified example.
The proposed solution involves applying L2-normalization exclusively to the K matrices while leaving Q matrices unnormalized. This design prevents K from dominating the training process while preserving Q's ability to modulate attention sharpness on a per-token basis through softmax temperature control. To validate their approach, the authors perform experiments training vision transformer models both with and without QUEST, demonstrating its effectiveness.

**Strengths:**

1) The paper addresses a significant issue affecting transformers across diverse applications and offers a straightforward solution to enhance their stability and performance.

2) The paper provides a thorough analysis of the attention mechanism and the function of its components. Claims regarding current limitations of Scaled Dot-Product Attention and proposed alternatives are validated through prior research and a toy example, which reinforce the paper's arguments and empirically demonstrate QUEST's effectiveness in addressing these issues.

3) QUEST's impact is illustrated through visualizations, which support the assertion that it prevents converging to less meaningfull attention weights.

4) The authors evaluate QUEST across multiple transformer architectures and tasks, demonstrating superior performance in nearly all experimental settings.

**Weaknesses:**

1) The main weakness of this work is that the reported improvements are relatively small, often around 1% or less. This raises some concern, especially since it appears that each experiment was run only once. Given the number of experiments conducted, it would be helpful to include a statistical summary of the results. Could the authors provide some form of statistical analysis of the improvements across tasks? Even a simple aggregate analysis could offer a clearer picture of the overall gains.

2) The paper assumes that the primary issue arises from the normalization of the K vectors, suggesting that Q should remain unnormalized during optimization. However, experiments involving only Q normalization also seem to yield positive effects. In the toy experiment, Q appears less stable than K but still more stable than the baseline, and in some cases even achieves better accuracy, particularly for certain learning rate values, where Q-Norm seems easier to optimize in the QUEST setup. This trend is also reflected in the ablation study, where Q-Norm improves over the baseline and performs comparably to K-Norm. Did the authors further investigate Q-Norm beyond the results presented in the paper?

**Questions:**

See Weaknesses

---

> ### Author Response · Authors · 2025-11-23
> **Response to Reviewer bMvw (part 1/2)**
>
> We thank the reviewer for their valuable feedback and suggestions to improve the paper. We address the weaknesses and questions raised by the reviewer below, in addition to the general rebuttal response.
>
> > W1. The main weakness of this work is that the reported improvements are relatively small, often around 1% or less. This raises some concern, especially since it appears that each experiment was run only once. Given the number of experiments conducted, it would be helpful to include a statistical summary of the results. Could the authors provide some form of statistical analysis of the improvements across tasks? Even a simple aggregate analysis could offer a clearer picture of the overall gains.
>
> We thank the reviewer for this suggestion to present statistical summaries to demonstrate the statistical significance of our results. We ran the ablation experiments in Table 1 three times and we provide the mean and standard deviations of these results in Table A1 in the revised paper. On running a 2-group t-test to check the statistical significance, we obtained p-values lower than 0.01 compared to the second best for all metrics in the table provided below.
>
> For one of the larger experiments, where we trained a ViT-Base model for 400+20 epochs using the DeIT-3 training recipe, we repeated the experiment without automatic mixed precision, to investigate if that further improves performance of the baseline. We found our results to be relatively stable (standard attention performance improved by 0.2% and QUEST performed similar).
>
> For the GraphGPS experiments in Tables 8 and 9, we already provided a statistical summary over 4 and 10 runs respectively. In the appendix, we also provide an experiment where we trained a ViT-Tiny model on different limited subsets of the ImageNet data (see Table A1). We provide the statistical summary for these results over 3 different data subsets. We find these experiments to validate the statistical significance of the results. While the performance improvements vary depending on the experiment, we would like to highlight the consistent improvement observed across a wide range of experiments.
>
> |  Attention |  Scaling | Top-1 (\%)  | Top-5 (\%)  |  NLL $\downarrow$ |
> |---|---|---|---|---|
> |  Standard | $1/\sqrt{D_h}$  |  $72.65\pm0.10$ |  $91.49\pm0.03$ |  $1.187\pm0.003$ |
> | QUEST  | -  |  $\textbf{73.36}\pm\textbf{0.16}$ |  $\textbf{91.86}\pm\textbf{0.06}$ |  $\textbf{1.160}\pm\textbf{0.004}$ |
> |  QNorm |  - |  $72.73\pm0.11$ | $91.44\pm0.02$  |  $1.188\pm0.004$ |
> |  QKNorm-HS | $^\dagger$ $\boldsymbol{C} \in \mathbb{R}^{L \times H}$  |  $72.49\pm0.10$ |  $91.43\pm0.09$ |  $1.198\pm0.009$ |
> |  QKNorm-DS |  $^\ddagger$ $\boldsymbol{C}_q, \boldsymbol{C}_k \in \mathbb{R}^{L \times D_h}$ |  $71.87\pm0.18$ | $90.96\pm0.11$  |   $1.231\pm0.008$ |

---

> ### Author Response · Authors · 2025-11-23
> **Response to Reviewer bMvw (part 2/2)**
>
> > W2. The paper assumes that the primary issue arises from the normalization of the K vectors, suggesting that Q should remain unnormalized during optimization. However, experiments involving only Q normalization also seem to yield positive effects. In the toy experiment, Q appears less stable than K but still more stable than the baseline, and in some cases even achieves better accuracy, particularly for certain learning rate values, where Q-Norm seems easier to optimize in the QUEST setup. This trend is also reflected in the ablation study, where Q-Norm improves over the baseline and performs comparably to K-Norm. Did the authors further investigate Q-Norm beyond the results presented in the paper?
>
> We have now added additional ablation experiments to evaluate QNorm further in other experiments. We evaluated both QNorm and QKNorm in the time series classification task and in the language modeling task (see Tables below). In time series classification, QNorm performs worse than standard attention. Here, training stability is not a concern and performance depends on whether the attention mechanism is able to learn useful patterns. In the language modeling experiment, QNorm performs slightly better than standard attention (similar to vision experiments) but underperforms QKNorm. Overall, on both these new ablations, we still find QUEST attention to perform best. In the general response to all reviewers, we provide an explanation for why QUEST could perform better than QNorm.
>
> **Language modeling ablation**
> (Cont. = contaminated, values reported are PPL)
>
> | Method | Model size (Parameters) | Attention | Clean-Val | Clean-Test | Cont. Test (1.5%) | Cont. Test (2.5%) | Cont. Test (5.0%) |
> |---|---|---|---|---|---|---|---|
> | Transformer | Medium (90M) | Standard | 27.441 | 28.851 | 36.234 | 40.866 | 55.012 |
> | Transformer | Medium (90M) | QUEST | **26.980** | **28.478** | **35.849** | **40.499** | **54.531** |
> | Transformer | Medium (90M) | QNorm | 27.233 | 28.688 | 36.262 | 40.853 | 55.451 |
> | Transformer | Medium (90M) | QKNorm-DS | 27.376 | 28.624 | 36.018 | 40.715 | 54.899 |
>
> **Time series classification ablation**
> (values reported are accuracies in %)
>
> | Dataset | Standard | QNorm | QKNorm-DS | QUEST |
> |---|---|---|---|---|
> | EthanolConcentration | 29.28 | 29.28 | 29.28 | 30.42 |
> | FaceDetection | 65.24 | 64.81 | 64.73 | 65.83 |
> | Handwriting | 42.00 | 41.06 | 49.18 | 49.18 |
> | Heartbeat | 77.56 | 75.61 | 77.56 | 78.54 |
> | JapaneseVowels | 98.38 | 98.92 | 98.38 | 98.38 |
> | PEMS-SF | 83.82 | 80.92 | 79.19 | 80.92 |
> | SelfRegulationSCP1 | 88.05 | 87.71 | 88.76 | 88.05 |
> | SelfRegulationSCP2 | 58.89 | 56.67 | 54.44 | 58.33 |
> | SpokenArabicDigits | 98.86 | 99.41 | 99.45 | 99.41 |
> | UWaveGestureLibrary | 86.88 | 85.00 | 86.81 | 86.25 |
> | Average | 72.90 | 71.94 | 72.78 | **73.53** |

---

### Official Review · Reviewer_MR6x · 2025-10-31

**Soundness:** 3
**Presentation:** 3
**Contribution:** 3
**Rating:** 4
**Confidence:** 3

**Summary:**

This paper revisits the instability issues in Transformer training, which arise from uncontrolled growth of query/key norms in standard softmax attention. The authors propose a simple yet effective variant, QUEST, that normalizes only the keys while keeping queries unnormalized. This breaks the mutual amplification between query and key norms, stabilizing training and allowing each query to control its own attention sharpness. The method is a drop-in replacement for standard attention and is evaluated across diverse domains. Experiments show that QUEST improves training stability, robustness to corruptions and adversarial attacks, and sometimes accuracy.

**Strengths:**

**Well-motivated**: Identifies a concrete cause of Transformer instability (growing norms) and proposes an intuitive fix requiring minimal code change.

**Strong Empirical Results**: Demonstrated consistent gains on multiple benchmarks (ImageNet, ADE20K, WikiText-103, GraphGPS, UEA datasets).

**Robustness Improvements**: Shows better performance under adversarial attacks (FGSM, PGD, AutoAttack) and data corruptions.

**General Applicability**: Works as a drop-in replacement across architectures (ViT, CrossViT, Transformer-XL, Graph Transformers).

**Theoretical & Empirical Insight**: Offers a helpful interpretation of attention norm dynamics and a controlled toy experiment demonstrating the effect of spurious correlations.

**Weaknesses:**

**Experiments:** Most experiments emphasize vision tasks; evaluations in NLP and other domains are relatively light.

**Novelty:** Normalizing keys is a simple modification; some may view it as an extension of prior QKNorm works rather than a fundamentally new paradigm.

**Ablation:** The comparison between normalizing only queries vs. only keys could be expanded.

**Analysis**:Theoretical analysis is mostly heuristic and lacks rigorous mathematical derivation or verified data support.

**Questions:**

1. Can the authors provide a more formal theoretical justification or analytical evidence beyond heuristic explanations?

2. In NLP tasks, does QUEST consistently outperform QKNorm or LayerNorm-based stabilizations?

3. More modality/field need to be test. Can QUEST be extended to multi-modal or diffusion Transformer architectures?

4. Previous [1] attributes training instability mainly to attention logit explosion and proposes QKNorm as a normalization-based remedy, how does QUEST theoretically or empirically differ from QKNorm in addressing this issue? Specifically, can the authors clarify whether QUEST’s key-only normalization provides comparable stability at large model scales, and if so, why this asymmetric normalization would outperform the symmetric approach proposed in [1]?

5. In NLP tasks, does QUEST consistently outperform QKNorm or LayerNorm-based stabilizations?


This is an interesting work, and I will raise my score if authors address all my concerns.

[1] Scaling Vision Transformers to 22 Billion Parameters

---

> ### Author Response · Authors · 2025-11-23
> **Response to Reviewer MR6x (part 1/2)**
>
> We thank the reviewer for their valuable feedback. We address the weaknesses and questions raised by the reviewer below, in addition to the general rebuttal response.
>
> > W1. Experiments: Most experiments emphasize vision tasks; evaluations in NLP and other domains are relatively light.
>
> Though we mainly focused on vision, we would like to highlight that we demonstrate the strength of the proposed QUEST attention across a broad range of experiments covering many different data modalities (vision, language, time series and graph structured data such as molecules). We have additionally evaluated robustness to corruptions and adversarial attacks (vision and language) and explainability. Further, we have demonstrated that this modification can also be combined with other improvements to attention through the experiments on Elliptical attention and CrossViT which uses cross-attention. We have attempted to cover multiple facets of evaluation, different data modalities, explored synergies with different transformer developments and also in the context of self-supervised pre-training.
>
> During this rebuttal phase, we explored one additional modality of point clouds. We conducted a point cloud segmentation experiment using the PointTransformer-V3 model from the Pointcept framework. We report the mIoU on the validation set of the nuScenes dataset. We found QUEST to perform better than standard and QKNorm attentions.
>
> | Model | Epochs | Attention | nuScenes val. mIoU (%) |
> |---|---|---|---|
> | PointTransformer-V3 | 100| Standard | 80.40 |
> | PointTransformer-V3 | 100| QUEST | 80.83 |
> | PointTransformer-V3 | 100| QKNorm-DS | 80.37 |
>
> > W2. Novelty: Normalizing keys is a simple modification; some may view it as an extension of prior QKNorm works rather than a fundamentally new paradigm.
>
> We agree with the reviewer that it is a simple modification but we consider this simplicity to be a strength of the proposed method, enabling it to be easily used as a drop-in replacement in attention layers across different data modalities and alongside other orthogonal Transformer improvements. The analysis of the query and key norms through which we explain the occurrence of the attention logit explosion and arrive at this simple modification bears novelty. Our work as well as QKNorm demonstrate that attention in the hyperspherical latent space can be a promising direction that is still underexplored. As we remark in the conclusion, aligning these models with optimization methods that are also geometrically aligned could be an interesting direction for future work. Furthermore, QKNorm only focused on vision transformers; however, we demonstrate broad applicability and synergies with other Transformer improvements (such as Elliptical attention and cross-attention paradigms in CrossViT).
>
> > W3. Ablation: The comparison between normalizing only queries vs. only keys could be expanded.
>
> We have conducted additional ablation experiments for both QNorm and QKNorm on the time series classification task and the language modeling task. Both these ablation experiments are in agreement with our initial observation that QUEST performs better than standard, QNorm and QKNorm attentions. We thank the reviewer for this suggestion to further strengthen our experimental results.
>
> | Dataset | Standard | QNorm | QKNorm-DS | QUEST |
> |---|---|---|---|---|
> | EthanolConcentration | 29.28 | 29.28 | 29.28 | 30.42 |
> | FaceDetection | 65.24 | 64.81 | 64.73 | 65.83 |
> | Handwriting | 42.00 | 41.06 | 49.18 | 49.18 |
> | Heartbeat | 77.56 | 75.61 | 77.56 | 78.54 |
> | JapaneseVowels | 98.38 | 98.92 | 98.38 | 98.38 |
> | PEMS-SF | 83.82 | 80.92 | 79.19 | 80.92 |
> | SelfRegulationSCP1 | 88.05 | 87.71 | 88.76 | 88.05 |
> | SelfRegulationSCP2 | 58.89 | 56.67 | 54.44 | 58.33 |
> | SpokenArabicDigits | 98.86 | 99.41 | 99.45 | 99.41 |
> | UWaveGestureLibrary | 86.88 | 85.00 | 86.81 | 86.25 |
> | Average | 72.90 | 71.94 | 72.78 | **73.53** |
>
> | Method | Model size (Parameters) | Attention | Clean-Val | Clean-Test | Cont. Test (1.5%) | Cont. Test (2.5%) | Cont. Test (5.0%) |
> |---|---|---|---|---|---|---|---|
> | Transformer | Medium (90M) | Standard | 27.441 | 28.851 | 36.234 | 40.866 | 55.012 |
> | Transformer | Medium (90M) | QUEST | **26.980** | **28.478** | **35.849** | **40.499** | **54.531** |
> | Transformer | Medium (90M) | QNorm | 27.233 | 28.688 | 36.262 | 40.853 | 55.451 |
> | Transformer | Medium (90M) | QKNorm-DS | 27.376 | 28.624 | 36.018 | 40.715 | 54.899 |
>
> > W4. Analysis: Theoretical analysis is mostly heuristic and lacks rigorous mathematical derivation or verified data support.
>
> > Q1. Can the authors provide a more formal theoretical justification or analytical evidence beyond heuristic explanations?
>
> See general response to all reviewers.

---

> ### Author Response · Authors · 2025-11-23
> **Response to Reviewer MR6x (part 2/2)**
>
> > Q2 and Q5. In NLP tasks, does QUEST consistently outperform QKNorm or LayerNorm-based stabilizations?
>
> The Transformer models that we consider in the language modeling task already use LayerNorms in the Transformer block. Our modification (as well as QKNorm) is only applied to the Attention layer and hence, does not replace the LayerNorm. We have added additional language modeling results evaluating QNorm and QKNorm on the Transformer-medium model. Both QNorm and QKNorm improve over standard attention in NLP but QUEST still performs marginally better than them.
>
> | Method | Model size (Parameters) | Attention | Clean-Val | Clean-Test | Cont. Test (1.5%) | Cont. Test (2.5%) | Cont. Test (5.0%) |
> |---|---|---|---|---|---|---|---|
> | Transformer | Medium (90M) | Standard | 27.441 | 28.851 | 36.234 | 40.866 | 55.012 |
> | Transformer | Medium (90M) | QUEST | **26.980** | **28.478** | **35.849** | **40.499** | **54.531** |
> | Transformer | Medium (90M) | QNorm | 27.233 | 28.688 | 36.262 | 40.853 | 55.451 |
> | Transformer | Medium (90M) | QKNorm-DS | 27.376 | 28.624 | 36.018 | 40.715 | 54.899 |
>
> > Q3. More modality/field need to be test. Can QUEST be extended to multi-modal or diffusion Transformer architectures?
>
> Repeating our answer to W1, we demonstrate the strength of the proposed QUEST attention across a broad range of experiments covering many different data modalities (vision, language, time series and graph structured data such as molecules). We have additionally evaluated robustness to corruptions and adversarial attacks (vision and language) and explainability. Further, we have demonstrated that this modification can also be combined with other improvements to attention through the experiments on Elliptical attention and CrossViT which uses cross-attention. We have attempted to cover multiple facets of evaluation, different data modalities and explored synergies with different transformer developments.
>
> Since QUEST is a simple drop-in replacement for attention, it can be easily added to multi-modal Transformers or Diffusion Transformers. We did not consider multi-modal or diffusion tasks in this work as they are significantly more compute expensive. For instance, multi-modal tasks in language/vision often rely on pre-trained models. Since we modify the attention mechanism, we would have to pre-train the models ourselves, which is compute expensive. As we noted earlier, during this rebuttal phase, we explored one additional modality of point clouds where QUEST still performed better than standard and QKNorm attentions.
>
> > Q4. Previous [1] attributes training instability mainly to attention logit explosion and proposes QKNorm as a normalization-based remedy, how does QUEST theoretically or empirically differ from QKNorm in addressing this issue? Specifically, can the authors clarify whether QUEST’s key-only normalization provides comparable stability at large model scales, and if so, why this asymmetric normalization would outperform the symmetric approach proposed in [1]?
>
> As we explained in page 3 of the paper, QKNorm $\ell_2$-normalizes both the queries and keys and then scales the queries and keys for all attention heads by the same scaling factor (which is a model parameter). This limits the expressivity of attention as it constrains all the tokens to have the same attention distribution sharpness (dependent on the fixed model parameter). Intuitively, this means that all tokens in all heads in a layer would attend to a similar number of tokens. On the other hand, QUEST allows each query to learn different query norms for each token (data dependent) and hence, control the sharpness of the attention distribution flexibly. This is the theoretical difference between QUEST and QKNorm.
>
> We experimented with a ViT-Huge model (600M parameters) and found similar observations as with other models. The biggest model that we could train on the available compute infrastructure with a reasonable batch size (=64) was a 2.4B parameter ViT model. We trained this model for 10 epochs using the already stable DeIT-3 training recipe and with a similarly large learning rate as [1]. We found that the model trained in a stable manner with QUEST attention and performed better than standard attention at the end of this short training. Note that training the 22B parameter model as in [1] was done on 1024 TPU v4 chips. We do not have the compute capability to perform such huge experiments.
>
> | Model | Parameters | Attention | Epochs | Top-1 | Top-5 | NLL $\downarrow$ |
> |---|---|---|---|---|---|---|
> |ViT-H/16 | 600M | Standard | 100 | 68.3 | 88.8 | 1.545 |
> |ViT-H/16 | 600M | QUEST | 100 | 69.2 | 89.4 | 1.495 |
> |ViT-H/16 | 600M | Standard | 100 + 20 | 76.2 | 93.3 | 1.042 |
> |ViT-H/16 | 600M | QUEST | 100 + 20 | 76.8 | 93.6 | 1.002 |
> |ViT-2B/16 | 2.4B | Standard | 10 | 3.1 | 9.4 | 6.153 |
> |ViT-2B/16 | 2.4B | QKNorm-DS | 10 | 5.2 | 14.9 | 5.740 |
> |ViT-2B/16 | 2.4B | QUEST | 10 | 11.9 | 27.8 | 5.011 |

---

> > ### Comment · Reviewer_MR6x · 2025-11-25
> > **Official Comment by Reviewer MR6x**
> >
> > Thank the authors for their response, which addresses part of my concerns. The remaining questions are as follows.
> >
> > **Q1.** I do not think that the ViT-H and ViT-2B experiments are fully converged given the limited number of training epochs. Considering the available computational resources, presenting only one such as ViT-H/16 results with fully converged training would be convincing enough for me. In addition, some concrete evidence or visualizations supporting the “similar observations” are needed.
> >
> > **Q2.** Prior studies on linear attention [1] have also identified the norm issues. It would be interesting to explore, in future work, whether QUEST can be applied in linear attention architectures such as [1, 2, 3] to mitigate the uncontrolled growth of query/key norms.
> >
> > [1] NaLaFormer: Norm-Aware Linear Attention for Transformer Models. arXiv preprint arXiv:2506.21137 (2025).
> >
> > [2] PolaFormer: Polarity-aware Linear Attention for Vision Transformers. ICLR 2025.
> >
> > [3] FLatten Transformer: Vision Transformer Using Focused Linear Attention. ICCV 2023.

---

> > > ### Author Response · Authors · 2025-12-02
> > > **Response to Reviewer MR6x**
> > >
> > > > Thank the authors for their response, which addresses part of my concerns. The remaining questions are as follows.
> > >
> > > We thank the reviewer for their initial feedback which enabled us to improve the paper and to strengthen our results. We are glad to see that part of their concerns have been addressed. We took some time to conduct longer trainings for the larger ViT models that have sufficiently converged (to the extent possible within our computational resources), in order to address their remaining concerns.
> > >
> > > > Q1. I do not think that the ViT-H and ViT-2B experiments are fully converged given the limited number of training epochs. Considering the available computational resources, presenting only one such as ViT-H/16 results with fully converged training would be convincing enough for me. In addition, some concrete evidence or visualizations supporting the “similar observations” are needed.
> > >
> > > Firstly, we would like to clarify what we meant by "similar observations" in our earlier response: “We experimented with a ViT-Huge model (600M parameters) and found similar observations as with other models.” Here, by “similar observations”, we meant that the proposed QUEST attention performed better than the considered baseline attention, consistent with what was reported in the paper for other ViT model sizes such as Small, Base and Large.
> > >
> > > Secondly, we would like to address the concern that the ViT-H training might not have converged. We have carried out longer trainings for 400+20 epochs for both ViT-L/16 and ViT-H/14 models, using the **DeIT-3** training recipe. The DeIT-3 training recipe consists of two phases - training with smaller input image resolution for 400 epochs and then finetuning with standard 224x224 image size for 20 epochs. Given the computational limitations, we made the following modifications to enable these longer trainings:
> > > - **ViT-L**: We used a smaller global batch size of 1024 instead of 2048 as in the original training setting.
> > > - **ViT-H**: We used a smaller global batch size of 1024 instead of 2048 as in the original training setting. We used a smaller input image size of 126x126 in the first training phase, instead of 182x182.
> > >
> > > For larger models, we found batch size reduction to have a negative impact on performance. This results in a reduced performance for ViT-L/16 compared to the results reported in DeIT-3 [2]. ViT-H/14 performance is reduced to a larger extent since we also trained with a smaller input resolution in the first training phase. Nevertheless, these experiments enable us to compare QUEST and standard attention in larger models that are trained until convergence. We observe that QUEST attention still results in slightly improved performance (+0.2%) compared to standard attention. Taking the results in Table 1 and our new experiments during the rebuttal into account, we find QUEST attention to improve performance *consistently* for all model sizes both in longer sufficiently coverged trainings as well as in shorter trainings with limited compute budget.
> > >
> > > | Model | Attention | Epochs | Top-1 Acc (%) |
> > > |---|---|---|---|
> > > | ViT-L/16 | Standard | 400+20 | 83.9 |
> > > | ViT-L/16 | QUEST | 400+20 | **84.1** |
> > > | ViT-H/14 | Standard | 400+20 | 83.2 |
> > > | ViT-H/14 | QUEST | 400+20 | **83.4** |
> > >
> > > [1] Touvron et al. "Training data-efficient image transformers & distillation through attention." ICML, 2021.
> > >
> > > [2] Touvron et al. "DeIT III: Revenge of the ViT." ECCV, 2022.
> > >
> > > > Q2. Prior studies on linear attention [3] have also identified the norm issues. It would be interesting to explore, in future work, whether QUEST can be applied in linear attention architectures such as [3, 4, 5] to mitigate the uncontrolled growth of query/key norms.
> > >
> > > We agree that exploring QUEST in the context of linear attention where similar training instabilities have been observed is an interesting direction for future work. We thank the reviewer for bringing these recent works on linear attention [3, 4, 5] to our notice. We will add them to our discussion in related work, where we discussed some earlier works related to linear attention.
> > >
> > > [3] NaLaFormer: Norm-Aware Linear Attention for Transformer Models. arXiv preprint arXiv:2506.21137 (2025).
> > >
> > > [4] PolaFormer: Polarity-aware Linear Attention for Vision Transformers. ICLR 2025.
> > >
> > > [5] FLatten Transformer: Vision Transformer Using Focused Linear Attention. ICCV 2023.

---

### Official Review · Reviewer_Bs2Y · 2025-11-01

**Soundness:** 3
**Presentation:** 4
**Contribution:** 3
**Rating:** 6
**Confidence:** 4

**Summary:**

For transformer architecture, in order to solve training instabilities when the norm of queries and keys arbitrarily increase, this paper propose  a new attention  (QUEST). This mechanism map the keys to a hyperspherical latent space and single token are still able to sharpen the attention distribution. Empirical experiments show the effectiveness of the proposed methods.

**Strengths:**

The paper is well-written and easy to follow. It has clear motivation and it identifies an attention instability that arises from growing norms of queries and keys. Based on that, this paper proposes an elegant solution driven by mathematical intuition and compare with relevant methods.

The proposed method is easy to be implemented.

This paper validate proposed method on broad experiments(vision, language, graph transformers and time-series.) and achieve superior performance. Moreover, in vision area, the explainability and robustness analysis are provided.

**Weaknesses:**

Training instability is not very proper since this paper does not analyse the optimization process but only after optimization.

Technical depth is limted since it is conceptual but not theoretical. The analysis is mostly based on observation and training instability is affected by the network Jacobian which is lack in the paper.

The scope is moderate

**Questions:**

See above

---

> ### Author Response · Authors · 2025-11-23
> **Response to Reviewer Bs2Y**
>
> We thank the reviewer for their valuable feedback. We address the weaknesses and questions raised by the reviewer below, in addition to the general rebuttal response.
>
> > W1. Training instability is not very proper since this paper does not analyse the optimization process but only after optimization.
>
> We did empirically analyze the optimization process with respect to the key and query norms which are central to our contribution. The key finding from this analysis is presented in Figure 4 in the main paper and in Figures A4 and A5 in the appendix. We found that a common failure case for standard and QNorm attentions involved the key norms of the biased answer token increasing as the training progressed (see Figure 4). However, these models do not distinguish between the keys of answer and non-answer tokens (see Figure A4 and A5 and discussion in section A3).
>
> For the ViT-Base model trained using the DeIT training recipe, we illustrate the progression of the key and query norms and the corresponding attention logits in Figure A6. The training instability caused by attention logit explosion is not a new finding but an explanation for why or how it occurs is still underexplored. Connecting this to model behavior in the presence of spurious correlations is a novel observation. Deeper theoretical analysis of the optimization dynamics is an interesting direction for future research, which we also remark in the conclusion.
>
> > W2. Technical depth is limited since it is conceptual but not theoretical. The analysis is mostly based on observation and training instability is affected by the network Jacobian which is lack in the paper.
>
> See general response to all reviewers. In this explanation, we do use the gradient updates to explain how instability can be avoided using QUEST (and also QNorm and QKNorm).
>
> > W3. The scope is moderate
>
> We demonstrate the strength of the proposed QUEST attention across a broad range of experiments covering many different data modalities (vision, language, time series and graph structured data such as molecules). We have additionally evaluated robustness to corruptions and adversarial attacks (vision and language) and explainability. Further, we have demonstrated that this modification can also be combined with other improvements to attention through the experiments on Elliptical attention and CrossViT which uses cross-attention. We have attempted to cover multiple facets of evaluation, different data modalities, explored synergies with different transformer developments and also in self-supervised pre-training. We observe consistent improvements across these wide range of experiments. This is a drop-in replacement for standard attention without any other changes to training setups. This simplicity further broadens the scope of applying this contribution.
>
> During this rebuttal phase, we explored one additional modality of point clouds. We conducted a point cloud segmentation experiment using the PointTransformer-V3 [1] model from the Pointcept framework. We report the mIoU on the validation set of the nuScenes dataset. We find QUEST to perform favorably compared to standard and QKNorm attentions.
>
> | Model | Epochs | Attention | nuScenes val. mIoU (%) |
> |---|---|---|---|
> | PointTransformer-V3 | 100| Standard | 80.40 |
> | PointTransformer-V3 | 100| QUEST | 80.83 |
> | PointTransformer-V3 | 100| QKNorm-DS | 80.37 |
>
> [1] Wu et al. Point transformer v3: Simpler faster stronger. CVPR 2024.

---

> > ### Comment · Reviewer_Bs2Y · 2025-11-26
> > **Response to Author**
> >
> > Thanks for authors' reply. After reading all other reviewers' comments. Although I think the scope is moderate, authors validate on extensive experiments. I still decide to maintain my scores (lean to acceptance).

---

### Official Review · Reviewer_5zrP · 2025-11-01

**Soundness:** 3
**Presentation:** 3
**Contribution:** 3
**Rating:** 6
**Confidence:** 4

**Summary:**

The authors propose a novel attention mechanism, QUEST (QUEry-modulated Spherical aTtention) to mitigate instability (attention collapse) during the training. The authors argue that the instabilities may come from spurious correlation, or stealing attention due to norm of a certain key. QUEST attempts to solve this by normalizing the key vectors. This coulb be interpreted between standard attention and QK-Norm attention, which focuses on the cosine similarity rather than the dot product. This paper demonstrates that QUEST stabilizes training and improves the performance through various experiment.

**Strengths:**

1. This paper clearly reformulates the problem. To calculate attention logit, which shows the relevance between query and key, the two obvious choices are  dot-product vs cosine similarity. The authors suggest a solution in between.

2. The authors present a broad experiments results.

3. Quest shows performance gain in IN-C experiment, which implies that QUEST can aggregate information from broad region.

**Weaknesses:**

1. toy example is too limite. Though the authors devote large portion of the paper into the toy example, the model does not have query, key projection head. The role of projection head is to extract information from embedding vectors. Analyzing attention without the learnable projection head is not compelling.

2. The theoretical analysis is not clear. The auithors argue that QUEST can  mitigate the trainig instability but does not show this metric with other method.

**Questions:**

1. Why does query normalization perform worse?
2.  Please clarify the elliptical-quest about scaling.
3. Please provide deeper comparison with QK-norm method.

---

> ### Author Response · Authors · 2025-11-23
> **Response to Reviewer 5zrP**
>
> We thank the reviewer for the valuable feedback. We address the weaknesses and questions raised by the reviewer below, in addition to the general rebuttal response.
>
> > W1. toy example is too limited. Though the authors devote large portion of the paper into the toy example, the model does not have query, key projection head. The role of projection head is to extract information from embedding vectors. Analyzing attention without the learnable projection head is not compelling.
>
> Thank you for pointing out this unclarity. In fact, we do use all the learnable projection heads in a standard Transformer. We used a simple one layer Transformer model in the toy example so that the roles of different components in the model can be easily interpreted. However, we use the standard implementation of the Transformer block.  In Algorithm A2 in the appendix, we showed the pseudo-code for the single Transformer model, that mimicked the standard Transformer block. But the Attention function in algorithm A1 was a simplified pseudo-code that was aimed at illustrating the change in QUEST attention compared to standard attention. We realize that this did not match with the Attention function call in line 9 of algorithm A2. We have addressed this by updating both algorithms A1 and A2 to align with each other. Additionally, we provide the actual code in the supplementary material (see file toy_example_transformer.py). We reiterate that we indeed use the linear projection to obtain the queries, keys and values, an output projection after the attention softmax and a further MLP projection in the Transformer block.
>
> > W2. The theoretical analysis is not clear. The authors argue that QUEST can mitigate the training instability but does not show this metric with other method.
>
> See general response to all reviewers.
>
> > Q1. Why does query normalization perform worse?
>
> See general response to all reviewers.
>
> > Q2. Please clarify the elliptical-quest about scaling.
>
> The goal with Elliptical-QUEST is to demonstrate the generality of QUEST and show that this modification can further boost the performance of Elliptical attention, which achieved state-of-the-art performance on adversarial robustness. Elliptical attention [1] only evaluated their approach on ViT-Tiny models and thus, we do not have optimal training setups to train Elliptical attention for larger ViT models. Hence, we have not evaluated Elliptical-QUEST with regards to scaling.
>
> [1] Nielsen et al. Elliptical attention. NeurIPS, 2024.
>
> > Q3. Please provide deeper comparison with QK-norm method.
>
> We have run additional experiments using both QNorm and QKNorm on the time series classification task and in the language modeling task. In both tasks, we still observe that QUEST attention performs favorably compared to QNorm and QKNorm (see Tables below). We thank the reviewer for this suggestion to further strengthen our experimental results. We will incorporate these results in the paper revision.
>
> **Language modeling ablation**
> (Cont.=contaminated, values reported are PPL)
>
> | Method | Model size (Parameters) | Attention | Clean-Val | Clean-Test | Cont. Test (1.5%) | Cont. Test (2.5%) | Cont. Test (5.0%) |
> |---|---|---|---|---|---|---|---|
> | Transformer | Medium (90M) | Standard | 27.441 | 28.851 | 36.234 | 40.866 | 55.012 |
> | Transformer | Medium (90M) | QUEST | **26.980** | **28.478** | **35.849** | **40.499** | **54.531** |
> | Transformer | Medium (90M) | QNorm | 27.233 | 28.688 | 36.262 | 40.853 | 55.451 |
> | Transformer | Medium (90M) | QKNorm-DS | 27.376 | 28.624 | 36.018 | 40.715 | 54.899 |
>
> **Time series classification ablation**
> (accuracies reported in %)
>
> | Dataset | Standard | QNorm | QKNorm-DS | QUEST |
> |---|---|---|---|---|
> | EthanolConcentration | 29.28 | 29.28 | 29.28 | 30.42 |
> | FaceDetection | 65.24 | 64.81 | 64.73 | 65.83 |
> | Handwriting | 42.00 | 41.06 | 49.18 | 49.18 |
> | Heartbeat | 77.56 | 75.61 | 77.56 | 78.54 |
> | JapaneseVowels | 98.38 | 98.92 | 98.38 | 98.38 |
> | PEMS-SF | 83.82 | 80.92 | 79.19 | 80.92 |
> | SelfRegulationSCP1 | 88.05 | 87.71 | 88.76 | 88.05 |
> | SelfRegulationSCP2 | 58.89 | 56.67 | 54.44 | 58.33 |
> | SpokenArabicDigits | 98.86 | 99.41 | 99.45 | 99.41 |
> | UWaveGestureLibrary | 86.88 | 85.00 | 86.81 | 86.25 |
> | Average | 72.90 | 71.94 | 72.78 | **73.53** |

---

### Author Response · Authors · 2025-11-23
**Theoretical justification about how QUEST improves the stability of training**

Reviewers 5zrP, MR6x and Bs2Y pointed to a lack of theoretical justification for the proposed QUEST attention. In the first paragraph of page 3 (lines 110-116), we provide an explanation for how a spurious correlation could trigger query and key norms to increase and result in an attention logit explosion, which is known to cause training instabilities. Here, we would like to expand on this explanation. In the context of Transformer gradient updates, [1] defines a reverse attention term as:
$$\tilde{E} = \Delta \hat{W}_o^T V^T \in \mathbb{R}^{N \times N}$$
$$R = A \odot \left( \tilde{E}^T - \mathrm{diag}(A\tilde{E}^T) \right)^T \sqrt{\frac{H}{D}} \in \mathbb{R}^{N \times N}$$

Here, $\hat{W_o}$ is the output projection weight, $V \in \mathbb{R}^{N \times D/H}$ and $\Delta \in \mathbb{R}^{N \times D}$ is the Vector Jacobian Products (VJPs) of $\hat{W_o}$. This term is then used to derive the VJPs for the query and key gradient updates (for token $j$) as:
$$\delta_q^j = R_j K \in \mathbb{R}^{D/H}$$
$$\delta_k^j = R_j^T Q \in \mathbb{R}^{D/H}$$


When attention, A concentrates on specific tokens, the reverse attention R also concentrates on those tokens (contributions from other tokens approach 0). The Vector Jacobian Products of the queries and keys are linear combinations of the keys and queries respectively. The tokens where attention is concentrated contribute significantly and the contributions from the other tokens are diminished. In the attention operation, higher key norms increase the attention towards that key token and reduce the attention to other key tokens. Hence, tokens with lower key norms and thereby lower attention probabilities, also contribute less to parameter updates.
When key norms of these tokens grow, this can consequently cause the query norms attending to that token to grow as well. This cross-play causes query and key norms to feed off each other and continue growing, resulting in an attention logit collapse. $\ell_2$-normalizing at least one of them can mitigate this effect. This can also be empirically observed in Figure A6 in the paper - the query norms and max logits in QUEST are stable and do not increase as in the case of standard attention. This is why QUEST, QNorm and QKNorm are all able to produce stable trainings. Note that, among these, QKNorm, is the only option that has previously been proposed in the literature [2,3] as a stabilizing modification of attention, to the best of our knowledge. QNorm and KNorm provide additional flexibility in the attention mechanism and hence perform better compared to QKNorm. QNorm, though stable, can still allow key norms of specific tokens to grow and hence, “steal attention” from other tokens. On the other hand, the query norms in QUEST can only influence the sharpness of attention and not which token should be attended to. We believe that this enables QUEST to learn better attention distributions as shown in the class-activation maps in the paper and perform robust under corruptions and adversarial attacks. We will elaborate on this justification and connection to existing theoretical results in the revised paper.

[1] Katz et al. Reversed attention: On the gradient descent of attention layers in GPT. In NAACL, 2025.

[2] Dehghani et al. Scaling vision transformers to 22 billion parameters. In ICML, 2023.

[3] Liu et al. Swin transformer v2: Scaling up capacity and resolution. In CVPR, 2022.

---

### Author Response · Authors · 2025-12-04
**Summary of paper revision**

We thank all the reviewers for their valuable and thoughtful feedback and for further engaging in rebuttal discussions. We have updated the paper with a revised version, incorporating the changes based on the rebuttal discussions. The changes made to the paper are highlighted with “RedOrange” font color. Here, we highlight some of the key updates.
___
### Theoretical analysis of QUEST
We added an analysis of the attention gradient updates that elaborates on the training instability observed in standard attention and sheds light on why QUEST performs better than other considered attention variants such as QKNorm and QNorm.
___

### Statistical significance of results
We conducted three independent runs of the image classification ablation experiment and reported the means and standard deviations of the results in Table A1. Additionally, we point to existing experiments which show standard deviations, such as training with different data subsets (section A.5.1; Table A2) and all experiments on graph transformers (section 4.3; Table 8 and 9). These experiments show that the improvements obtained using QUEST compared to the baselines are statistically significant.
___
### Added comparisons with QKNorm and QNorm
We have added ablation experiments comparing QUEST to QKNorm and QNorm in the time series experiment and the language modeling experiment, to further strengthen our observations regarding these comparisons across modalities.
___
### Broad applicability across different modalities
We added an additional experiment of pointcloud segmentation using the PointTransformer-V3 model (see section A.5.7; Table A9) to add one more modality to our extensive experiments, demonstrating broad applicability of QUEST.
___
### Large-scale ViT models evaluated in image classification
We have evaluated larger models such as ViT-Large and ViT-Huge using DeIT-3 training recipe with a larger number of training epochs, to ensure convergence (see Table 2). We already conducted longer experiments on ViT-Tiny and ViT-Base, that were trained for longer number of epochs. These experiments show that QUEST attention indeed performs favorably in longer trainings. We also showed a short experiment on ViT-2B (a 2 Billion parameter model) in Table A6 where QUEST seemed to learn significanlty faster than the baselines. A 2 Billion parameter model was the largest model that we could feasibly train.
___

---

### Meta-Review · Area_Chair_876e · 2026-01-07

**Summary:**

The authors proposed a simple attention mechanism called QUEST, where instead of the standard softmax attention over unnormalized queries and keys, they constrain all keys to lie on a hypersphere (L2-norm normalized) while keeping query norms free so that each query can still control how sharp or focused its attention distribution is. This aims to prevent key norms from arbitrarily growing and causing attention instability or spurious focus on irrelevant tokens.

- A recurring concern is that key/query normalization is not new; QUEST can be viewed as the variant that has been studied in the Q/Knorm paper. Reviewers asked for deeper comparisons to QKNorm and a clearer articulation of the mechanism-level distinction (especially why key-only normalization is preferable to query-only or symmetric QK normalization).
- Several reviewers also questioned about the experimental results regarding low performance of baselines and training epochs. Most of concerns have been addressed in the authors' response.

**Reviewer Concerns:**

one reviewer (5T4g) remains strongly unconvinced, focusing on baseline strength, convergence, and interpretability evidence.
- 5T4g found the Figure 1 narrative confusing/unconvincing (model attends to correct region but predicts wrong label; unclear advantage of QUEST). Even after rebuttal, this reviewer maintained that the figure doesn’t convincingly support claims and may confuse. AC personally thinks the figure is made for illustration purpose thus there is no need to put heavy explanations in the beginning of this paper.
- Regarding the training convergence, authors later ran longer converged DeIT-3 style 400+20 for ViT-L/16 and ViT-H/14, showing consistent +0.2% gains with QUEST vs standard under their constrained batch/resolution setup.

Overall, the method is extremely simple, broadly applicable, and the empirical story is surprisingly consistent across modalities; multiple reviewers rate it 6 (above threshold) with good soundness/presentation. MR6x and 5zrP see it as well-motivated and generally effective; Bs2Y explicitly “leans to acceptance” after rebuttal; bMvw views it as addressing an important issue with strong analysis and broad experiments. The rebuttal directly addressed the most actionable concerns: added QNorm/QKNorm ablations in language/time-series, added longer converged larger-ViT runs, and provided statistical summaries. With these, the remaining objections are mostly about “how much is enough” for scaling and novelty, rather than clear flaws. For ICLR, a clean, drop-in attention variant with consistent gains and improved stability/robustness across domains is typically considered publishable, albeit as a borderline acceptance.

**Reviewer Scores:**

From the multiple rounds of discussions between authors and reviewer MR6x, most questions have been answered. It is likely that the scores will be bumped up to 6.

---

### Decision · Program_Chairs · 2026-01-26

Accept (Poster)